# *Eurotium cristatum* Ameliorates Glucolipid Metabolic Dysfunction of Obese Mice in Association with Regulating Intestinal Gluconeogenesis and Microbiome

**DOI:** 10.3390/foods14244273

**Published:** 2025-12-12

**Authors:** Weirong Yang, Ning Han, Xiangnan Zhang

**Affiliations:** 1Faculty of Science, The University of Technology Sydney, Sydney 2007, Australia; amy.yang2001@hotmail.com; 2College of Food Engineering and Nutritional Science, Shaanxi Normal University, Xi’an 710119, China; hanning@snnu.edu.cn

**Keywords:** *Eurotium cristatum*, glucolipid metabolism, gut microbiota, intestinal gluconeogenesis, linoleic acid metabolism

## Abstract

*Eurotium cristatum* (EC), a fungus derived from Fu brick tea, exhibits anti-obesity potential, but its mechanisms regulating intestinal gluconeogenesis (IGN) remain unclear. This study aimed to elucidate whether EC alleviates obesity and glucolipid metabolic disorders by modulating the gut microbiota and activating the IGN pathway. The 8-week EC administration at low (10^4^ CFU/mL), medium (10^6^ CFU/mL), and high doses (10^8^ CFU/mL) ameliorated high-fat-diet (HFD)-induced metabolic abnormalities, including aberrant weight gain, dyslipidemia, glucose intolerance and hepatic injury with effects showing a dose-dependent trend. EC treatment significantly activated IGN, as indicated by increased colonic levels of short-chain fatty acids (SCFAs) and succinate (key IGN substrates) and the upregulation of IGN-key enzymes (PEPCK, FBPase, and G6Pase). In addition, EC treatment significantly alleviated the HFD-induced gut dysbiosis by reducing the *Firmicutes*/*Bacteroidetes* ratio and enriching beneficial bacteria such as *Lachnospiraece_NK4A136_group*, *Bacteroidota* and *Alloprevotella*. Non-targeted metabolomics analysis revealed that EC significantly altered the linoleic acid metabolism, specifically decreasing the relative levels of bile acid and chenodeoxycholic acid (*p* < 0.01) while increasing those of linoleic acid and ricinoleic acid (*p* < 0.05). EC treatment reshaped the gut microbiome, promoted the production of beneficial metabolites (e.g., SCFAs), and consequently activated the IGN pathway, ultimately ameliorating host glucose and lipid metabolic disorders. Our findings provide mechanistic insights into the anti-obesity effects of EC, suggesting its potential for further investigation as a dietary intervention for metabolic diseases.

## 1. Introduction

The prevalence of obesity or glucolipid metabolism disorder has reached epidemic proportions worldwide, posing a significant public health challenge [1,2]. Among the various dietary patterns contributing to obesity, the Western-style diet, characterized by its high-fat diet (HFD), has emerged as a major contributor [2]. Obesity and glucolipid metabolic disorders are widely acknowledged as significant risk factors for the development and progression of various health conditions, including diabetes, liver diseases, cardiovascular diseases, and even certain types of cancers [3,4,5]. Changes in lifestyle, dietary system, medications, and surgery are implemented in clinics for managing body weight and glucolipid metabolic disorders [6]. Despite significant efforts dedicated to combating obesity and overweight, the prevalence of obesity and glucolipid metabolic disorders continues to rise [7]. Therefore, there is an urgent need for developing novel dietary agents and strategies to effectively address obesity and metabolic disorders.

Notably, recent studies have highlighted the pivotal role of gut microbiota in the onset of obesity [5]. Several studies have shown that obesity is associated with a reduction in beneficial bacteria, such as *Bifidobacteria* and *Lactobacilli*, and an increase in potentially harmful bacteria, such as certain species of *Firmicutes* and *Proteobacteria* [8,9]. Characterized by an imbalance in microbial communities (gut dysbiosis) and a consequent decline in beneficial metabolites such as short-chain fatty acids (SCFAs), a disrupted gut ecosystem is strongly implicated in the development of obesity [10]. Published papers have indicated that the SCFAs can activate the induction of intestinal gluconeogenesis (IGN), inducing beneficial effects on glucose and energy homeostasis [11]. Specifically, SCFAs stimulate glucose production in the gut by activating key enzymes involved in IGN, such as fructose-1,6-bisphosphatase (FBPase) and glucose-6-phosphatase (G6Pase) [12]. The glucose released through IGN is detected by a glucose sensor in the portal vein and transmitted to the brain via the peripheral nervous system, leading to an inhibition on glucose release in the liver to maintain stable blood sugar levels [13]. In this regard, this novel indirect strategy which can restore the dynamic balance of gut microbiota and subsequently activating IGN have thus been proposed to prevent and treat obesity.

Recently, fungi have received considerable attention due to their potential health-promoting effects, especially their anti-obesity effect [14]. *Eurotium cristatum* (EC) is a predominant probiotic fungus that undergoes a unique “flowering” process and is traditionally used in the manufacturing of Fu brick tea (FBT) [15,16]. EC can secrete multiple active metabolites during the process of “flowering”, which is non-toxic and safe [17]. Studies indicate that FBT exerts protective effects against various metabolic disorders through multiple mechanisms, such as alleviating inflammation, regulating blood lipids and glucose, modifying gut microbial composition, and facilitating weight reduction [18,19,20]. However, it remains unclear whether *Eurotium cristatum* (EC), the predominant probiotic fungus contributing to the characteristic quality of FBT, is directly responsible for the tea’s reported anti-obesity effects. Our hypothesis posits that the anti-obesity properties of FBT are mediated by its resident fungus, EC. This could occur either through the survival and activity of live EC in the gut, or via the gut microbiota-modulating capacity of non-viable EC cells, ultimately leading to the observed metabolic benefits.

In this study, we systematically elucidated the mechanism by which EC ameliorates obesity through the “gut microbiota–SCFA–IGN” axis. For the first time, we demonstrated that EC could modulate the gut microbiota, increase colonic SCFAs levels, and regulate linoleic acid metabolism in obese mice, thereby enhancing IGN and significantly attenuating obesity. Our findings suggest that EC, as a probiotic, exhibits considerable potential for obesity prevention and treatment. This research is highly relevant to the global audience of Foods as it identifies EC as a promising fungal probiotic and functional food ingredient. The findings offer a scientific foundation for developing novel, microbiota-targeting strategies against the worldwide epidemic of obesity and metabolic syndrome. This aligns with the growing consumer and industrial interest in natural, food-derived solutions for health maintenance, moving beyond traditional tea consumption to the utilization of its defined, bioactive constituents.

## 2. Materials and Methods

### 2.1. Materials and Reagents

*Eurotium cristatum* (EC) was obtained from the China Center of Industrial Culture Collection (Beijing, China). The lyophilized powder was initially activated and cultured on Potato Dextrose Agar (PDA) plates at 28 °C for 7 days. A single colony was then inoculated into Potato Dextrose Broth (PDB) and incubated in a shaker at 28 °C and 150 rpm for 5 days to obtain the primary seed culture. For large-scale preparation, the primary seed culture was transferred to a fermenter containing sterile PDB (1% *v*/*v* inoculation) and fermented under the same conditions. The resulting fungal biomass was harvested via centrifugation (4 °C, 8000× *g* for 10 min), washed twice with sterile physiological saline (0.9% NaCl), and re-suspended in saline. The concentration of the live EC suspension was determined by plating serial dilutions on PDA plates and counting the colony-forming units (CFU) after incubation at 28 °C for 48–72 h. Crucially, to confirm the stability of the suspension under conditions mimicking the gavage procedure, we conducted a preliminary stability test. The prepared suspensions at all three target concentrations (10^4^, 10^6^, and 10^8^ CFU/mL) were held at 4 °C (the temperature used for short-term storage during the daily gavage period) and sampled for CFU counting at 0, 2, and 4 h. No significant decrease in viable count (*p* > 0.05) was observed within this 4 h window, confirming the stability of the suspension for the duration of the daily gavage process. Commercial assay kits were used for the measurement of serum lipids (TC, TG, LDL-C, HDL-C) and hormones (leptin, adiponectin). Lipid assay kits were sourced from Huili Biotechnology Co., Ltd. (Changchun, China), while ELISA kits for leptin (LEP) and adiponectin (ADPN) were obtained from Jiangsu Meimian Industrial Co., Ltd. (Yancheng, China). Reagents for biochemical assays (AST, ALT, leptin, adiponectin) and chemical standards (succinate, acetic acid, propionic acid, butyric acid) were sourced from Nanjing Jiancheng Bioengineering Institute (Nanjing, China) and Sigma (St. Louis, MO, USA), respectively. Primary antibodies for Western blotting—including UCP1, SGLT3, PEPCK, FBPase, and G6Pase—were sourced from Proteintech Group, Inc. (Wuhan, China). All chemicals used were analytical grade and commercially obtained.

### 2.2. Animal Experimentation

Male C57BL/6J mice (four weeks old) were sourced from the Experimental Animal Research Center of Shaanxi Normal University. After acclimatization to the laboratory environment for one week, the mice were randomly assigned to one of five experimental groups, with eight animals per group. Group assignment was performed using a computer-generated randomization table to ensure unbiased and reproducible distribution across experimental conditions: normal control (NC) group, high-fat-diet (HFD) control group, 10^4^ CFU mL^−1^ EC treatment (EC-L) group, 10^6^ CFU mL^−1^ EC treatment (EC-M) group and 10^8^ CFU mL^−1^ EC treatment (EC-H) group. Dietary regimens differed between groups: NC mice received standard chow, whereas all other groups were provided with a HFD. Meanwhile, mice in the NC and HFD groups were given physiological saline by gavage (i.g., 0.2 mL). To ensure dosing consistency, the EC suspensions were prepared fresh daily from the same master stock, and the CFU count was confirmed before each gavage. Mice in the EC-L, EC-M, and EC-H groups received a daily oral gavage of 0.2 mL of the corresponding EC suspension, delivering consistent doses of approximately 10^4^, 10^6^, and 10^8^ CFU per mouse per day, respectively. Throughout the intervention period, and during sample collection and outcome assessments (including biochemical analyses, histopathological evaluation, OGTT/ITTs, and Western blot analysis), the investigators involved in data collection and analysis were blinded to the group allocations. The blinding was maintained until all data analyses were completed. Upon completion of the 8-week study, mice were sacrificed. Samples of serum, liver, adipose tissue, and intestinal contents were collected for analysis and maintained at −80 °C until processing.

### 2.3. Biochemical Measurements

Key serum parameters were assessed as follows: TC, TG, LDL-C, and HDL-C levels were measured using corresponding enzymatic colorimetric kits (employing CHOD-PAP, GPO-PAP, and homogeneous methods, respectively) according to the manufacturer’s protocols. Meanwhile, adiponectin and leptin concentrations were quantified using specific ELISA kits per the provided instructions. The assays are based on the sandwich ELISA principle, where captured antibodies specific to each analyte are coated onto the microplate. The intensity of the colorimetric signal, measured at 450 nm, is proportional to the concentration of the target hormone in the sample. A standard curve was generated for each assay to calculate the precise concentrations. The activities of AST and ALT in liver homogenates were determined using commercial assay kits following the manufacturer’s protocols. These assays are based on the principle that AST and ALT catalyze specific reactions involving α-ketoglutarate and L-aspartate or L-alanine, respectively, leading to the formation of NADH, which is monitored by the increase in absorbance at 340 nm.

### 2.4. Histopathological Examination and Immunofluorescence

Histopathological analysis was conducted on adipose and liver tissues fixed in 4% neutral formalin. Fixed tissues were paraffin-embedded, sectioned, and stained with H&E and Oil Red O for microscopic observation (Olympus, Japan). For immunofluorescence, after antigen retrieval, tissue sections were blocked with 5% goat serum and incubated with primary antibody (UCP1, 1:500) at 4 °C overnight. Following a standard immunofluorescence staining protocol [21], images were acquired using a ZEISS Axio Imager M2 (Germany) upright microscope. For analysis, all tissue sections were coded prior to evaluation to ensure the pathologist was blinded to the group allocations. The blinding procedure was maintained until all analyses and scoring were completed.

### 2.5. Oral Glucose Tolerance Test (OGTT) and Insulin Sensitivity Test (ITT)

The OGTT and ITT were carried out one week prior to the termination of the experiment. Mice were subjected to an 8 h fast before receiving either an oral glucose load (2.0 g/kg body weight) for the OGTT or an intraperitoneal injection of insulin (0.75 U/kg body weight) for the ITT. Blood glucose concentrations were obtained from tail-tip samples at 0, 30, 60, 90, and 120 min following glucose or insulin administration. These time-course measurements were used to evaluate glucose clearance efficiency, insulin sensitivity, and overall metabolic regulation in response to the experimental treatment.

### 2.6. Western Blot Analysis

Western blot methods used in this study refer to previous research [22]. Following homogenization of colon tissues in RIPA buffer and centrifugation (10,000× *g*, 8 min, 4 °C), total protein in the supernatant was quantified using a BCA assay. Protein samples, denatured in loading buffer at 100 °C for 10 min, were loaded equally across lanes. Electrophoresis was performed using 10% stain-free gels (Bio-Rad), followed by transfer onto PVDF membranes. The membranes were then incubated in 5% non-fat milk blocking solution. After UV activation, the total protein in the stain-free gel was visualized to confirm equal loading and uniform transfer prior to membrane blocking. Primary antibody incubation was carried out at 4 °C overnight. After washing, the membranes were subjected to a 1 h incubation with the relevant secondary antibody at room temperature. All antibodies were validated by the manufacturer using knockout/knockdown cell lines or recombinant protein, and the observed molecular weights in our blots were consistent with the expected sizes for the target proteins. Immunoreactive proteins were captured using a ChemiDoc imagining system. The gray values of the protein bands were calculated using ImageJ software (1.53e).

### 2.7. Gut Microbiota Analysis

Genomic DNA of the gut microbiota was first isolated from colonic contents with the E.Z.N.A. Soil DNA Kit (Omega Bio-tek, Norcross, GA, USA). The isolated DNA served as the template for amplifying the 16S rRNA gene V3-V4 regions via PCR on an ABI GeneAmp 9700 system (Waltham, MA, USA), employing the primer pair 338F/806R. Each sample was assigned a unique 8-base barcode during this amplification step [23].

### 2.8. Analysis of SCFAs and Succinate in Colonic Contents

To quantify short-chain fatty acids (acetic, propionic, and butyric acids) and succinate, colonic content samples from each mouse were analyzed by GC-MS [24].

### 2.9. Untargeted Metabolomics Analysis

To prepare samples for non-targeted metabolomics, colonic contents (100 mg) were extracted with a ternary solvent system (water, methanol, acetonitrile, 1:1:1, *v*/*v*). Following homogenization and centrifugation, the supernatant was diluted, filtered (0.22 μm PVDF), and stored at −80 °C. This analytical approach employed UPLC-QTOF-MS/MS (Thermo Fisher Scientific, Waltham, MA, USA) [25]. The acquired data were used to pinpoint differential metabolites and to explore enriched metabolic pathways through the KEGG database and MetaboAnalyst 6.0 tools.

### 2.10. Statistical Analysis

Statistical significance was evaluated using appropriate tests for each data type. One-way ANOVA (Tukey’s post hoc), two-way repeated-measures ANOVA (Sidak’s test), and FDR correction were applied to biochemical/Western blot, OGTT/ITT, and omics datasets, respectively. A *p*-value < 0.05 after FDR adjustment defined significance for omics data. All values are reported as mean ± SD from triplicate measurements.

## 3. Results

### 3.1. EC Inhibited Obesity in HFD Mice

As shown in Figure 1, HFD remarkably increased the body weight and weights of the epididymal, subcutaneous and retroperitoneal fat in comparison with NC mice, respectively (*p* < 0.01). Furthermore, no significant difference in food intake was observed among the five groups of mice (Appendix A). The EC intervention effectively suppressed the HFD-induced increases in the body and liver weight, and contributed to significant reductions in epididymal, subcutaneous and retroperitoneal fats in HFD mice (*p* < 0.05). Meanwhile, the HFD mice showed a decrease in the brown fat (Figure 1G, *p* > 0.05) and an increase in the hepatosomatic index (HI) (Figure 1C, *p* > 0.05) and fat index (FI, *p* < 0.01, Figure 1H), when compared to the NC mice. Oral EC supplementation counteracted the significant increases in HI and FI induced by HFD (*p* > 0.05). Furthermore, the treatment also mitigated the HFD-associated decrease in brown fat mass, albeit not significantly in this model (*p* > 0.05). In addition, H&E staining of epididymal and subcutaneous adipose in HFD mice showed a significant increase in the size of adipocyte cells, while the oral administration of EC at 10^4^, 10^6^ and 10^8^ CFU mL^−1^ obviously decreased the adipocyte size of HFD mice (Figure 1I).

### 3.2. EC Alleviated Abnormal Glucolipid Metabolism and Liver Injury While Regulating Glucose Homeostasis in HFD Mice

In contrast to NC mice, HFD-fed animals exhibited a severely disrupted serum lipid profile (Figure 2A–D). This was evidenced by considerably increased levels of TC, TG, and LDL-C, alongside a notable decline in HDL-C (all *p* < 0.01). Interestingly, compared with the mice in HFD group, EC-H reduced the levels of serum TC, TG, and LDL-C of mice by 50.2%, 58.1% and 53.9%, respectively (*p* < 0.01), while serum HDL-C level of EC-H-treated mice was elevated by 45.9% (Figure 2C, *p* < 0.01). As shown in Figure 2E,F, compared to the NC mice, the serum levels of adiponectin (*p* < 0.01) and leptin (*p* > 0.05) were decreased in HFD mice. EC treatment significantly up-regulated adiponectin (*p* < 0.01) and leptin (*p* < 0.01) in the serum of HFD mice. Moreover, the hepatic AST and ALT activities in EC-H treated mice were reduced by 42.1% (*p* < 0.01) and 48.4% (*p* < 0.01), relative to the HFD mice, respectively (Figure 2G,H), suggesting that EC exhibited the protective effects against HFD-induced liver injury in mice.

Furthermore, histological analyses using Oil Red O and H&E staining (Figure 2I) were conducted to corroborate the biochemical findings. Oil Red O-stained liver sections showed that mice in the NC group displayed normal hepatic morphology with minimal lipid deposition, whereas HFD-fed mice exhibited extensive accumulation of lipid droplets within hepatocytes. Notably, EC supplementation markedly reduced both the number and size of hepatic lipid droplets in obese mice.

Similarly, H&E-stained liver sections from HFD-fed mice revealed pronounced hepatic steatosis, including ballooned, lipid-laden hepatocytes, severe cytoplasmic degeneration, and disrupted cellular architecture when compared with NC mice (Figure 2I). In contrast, liver tissues from EC-treated mice demonstrated preserved cytoplasmic integrity, clearly defined cell boundaries, and distinct nuclei and nucleoli, closely resembling the normal morphology observed in the NC group. These findings indicate that EC supplementation substantially mitigated HFD-induced hepatic injury.

We assessed the effects of EC on glucose tolerance and insulin sensitivity by calculating the area under the curve (AUC) for both the OGTT and ITT. As shown in Figure 3A,B, the blood glucose levels of EC-treated mice were generally lower than those of HFD mice at most time points during the OGTT, although this difference did not reach statistical significance across the entire curve (*p* > 0.05). A significant decrease in blood glucose levels following insulin administration was observed in all EC dosage groups (EC-L, EC-M, EC-H) relative to the HFD group (all *p* < 0.01) (Figure 3C,D). Taken together, these results suggested that EC significantly improved the glucose tolerance and insulin sensitivity of obesity mice.

### 3.3. EC Activated Intestinal Gluconeogenesis (IGN) and Adipose Thermogenesis

Recent evidence suggests that activating IGN plays an important role in improving glucose homeostasis [11]. Thereafter, we determined the expression levels of colonic gluconeogenesis proteins (SGLT3, PEPCK, FBPase and G6Pase) of mice by Western blotting analysis, and the results are shown in Figure 3E. In comparison with NC mice, the expressions of colonic SGLT3, PEPCK, FBPase and G6Pase in HFD mice were down-regulated, while EC treatment dramatically increased their expressions (Figure 3F–I), suggesting that EC may improve blood glucose balance in HFD mice by regulating IGN. In addition, to investigate whether the administration of EC may activate the thermogenic activity of brown adipose tissue (BAT), the thermogenic progress marker protein UCP1 was determined by immunofluorescence (IF). IF results showed that long-term HFD feeding severely inhibited the expression of UCP1 in BAT of mice as compared with NC feeding, while EC treatment significantly increased the UCP1 expression in HFD mice (Figure 3J), indicating that EC can promote BAT thermogenesis in HFD fed mice.

### 3.4. EC Reshaped the Gut Microbiota Composition of HFD Mice

The profiling of the colonic microbiota via 16S rRNA gene (V3–V4) pyrosequencing demonstrated distinct microbial communities between EC and HFD groups at the OTU level (Figure 4A). The number of OTUs unique to each group was quantified: 151 in NC mice, 97 in HFD mice, and 114, 103, and 126 in the EC-L, EC-M, and EC-H groups, respectively. In addition, alterations in microbial community richness were assessed using the Chao1 index. As shown in Figure 4B, High-fat-diet (HFD) intervention markedly reduced the Chao1 index compared with the NC group, indicating a decline in microbial richness. In contrast, EC supplementation significantly restored and enhanced this index. To further characterize community-level differences, β-diversity analyses—principal coordinates analysis (PCoA) and non-metric multidimensional scaling (NMDS)—were performed, providing a comprehensive overview of how HFD and EC treatments shaped the overall composition and structural configuration of the gut microbiota. (Figure 4C,D). Consistent with the PCoA and NMDS results, hierarchical cluster analysis also showed distinct groupings for all five experimental conditions (Figure 4E). Figure 4F displays a bar plot of the relative abundances at the phylum level, highlighting the top eight taxa across groups. EC treatment induced a pronounced shift in the gut microbial community of HFD-fed mice, characterized by an increased relative abundance of Bacteroidota and reduced abundances of Firmicutes and Proteobacteria (Figure 4G–J). As illustrated in Figure 4I, the *Firmicutes*-to-*Bacteroridetes* ratio (*F*/*B* ratio) in the HFD group was significantly higher than that in the NC group (*p* < 0.01), whereas the administration of EC significantly lowered the *F*/*B* ratio of HFD mice (*p* < 0.01).

Compared to NC mice, HFD altered the genus-level composition, elevating *Helicobacter* and *Faecalibaculum* and reducing *Alloprevotella* and *Lachnospiraece_NK4A136_group* (Figure 5A). Treatment with EC-H reversed these trends: it decreased the former two genera by 31.4% (*p* > 0.05) and 65.8% (*p* < 0.05); and increased the latter two by 3.57-fold and 20.0-fold (both *p* < 0.05), respectively, compared to the HFD group (Figure 5B–E). The numbers of differentially abundant bacterial clades (LDA > 3.0), as determined by linear discriminant analysis effect size (LEfSe), were 5, 6, 6, 4, and 4 for the NC, HFD, EC-L, EC-M, and EC-H groups, respectively (Figure 5F,G). The most discriminative features at the taxonomic level are shown in Figure 5G. Under HFD conditions, the family *Leuconostoc* and genus *Lactococcus* were most affected, while following EC administration, the genera *Allobaculum* and *Faecalibaculum* exhibited the greatest changes. In contrast to the marked changes observed in other genera, the relative abundances of *Eggerthellaceae* and *Ruminococcus* were not significantly different between the HFD and EC-treated groups (*p* > 0.05).

### 3.5. EC Promotes Colonic SCFAs and Succinate Production and Alleviates Linoleic Acid Metabolism Dysregulation in HFD-Fed Mice

EC-H intervention reversed the HFD-induced decline in total SCFAs (*p* < 0.01 vs. NC), achieving a 37.3% increase (*p* < 0.05, Figure 6A). This was reflected in elevated concentrations of acetic, propionic, and butyric acids, as well as succinate (Figure 6B–E). These SCFA profiles were significantly correlated with the abundance of specific gut microbes (positively with *Bacteroides*, *Lactobacillus*, and *Odoribacter*; negatively with *Helicobacter* and *Faecalibaculum*; *p* < 0.05, Figure 6F).

We further systematically analyzed the impact of EC-H on the metabolic profile of colonic contents in HFD-fed mice using UPLC-QTOF-MS technology. Multivariate statistical analysis (PLS-DA/OPLS-DA) revealed that the metabolic characteristics of EC-H group were significantly separated from the HFD group and more closely resembled the NC group levels (Figure 7A,B), indicating that EC effectively reversed HFD-induced metabolic disorders. Among the 70 identified differential metabolites (VIP > 1.0), 18 were annotated as human homologs. Heatmap analysis further confirmed the high similarity between the metabolic profiles of EC-H and NC groups (Figure 7C). KEGG pathway enrichment analysis demonstrated that these metabolites were primarily involved in key metabolic pathways, with linoleic acid metabolism showing the highest impact value, followed by arginine biosynthesis (Figure 7D). Specifically, compared with the HFD group (Figure 7E–H), the EC-H group showed significantly increased levels of linoleic acid and ricinoleic acid (*p* < 0.05), while the levels of cholic acid and chenodeoxycholic acid were significantly decreased (*p* < 0.01).

Furthermore, the expression levels of key metabolites (triethylphosphate, cholic acid, etc.) in the EC-H group significantly approached those in the NC group (Figure 8A,B). Notably, the expression differences of 18 metabolites between EC-H and HFD groups (*p* < 0.01) were significantly greater than those between EC-H and NC groups. Further analysis also revealed that metabolites such as pterin, triethylphosphate, and ureidopropionate were positively correlated (*p* < 0.05) with the relative abundance of gut microbiota including *Desulfovibrio* and *Lachnospiraceae UCG-006*. In contrast, metabolites such as bezafibrate, pterin, 4-O-methylphloracetophenone, dodecanedioic acid, ketoleucine, triethylphosphate, and ureidopropionate showed negative correlations (*p* < 0.05) with the relative abundance of *Faecalibaculum* and *Bifidobacterium*. These results demonstrate that EC ameliorates obesity-associated metabolic disorders by modulating gut microbiota composition, promoting the production of SCFAs and succinate, suppressing abnormal bile acid accumulation, and activating beneficial metabolic pathways such as linoleic acid metabolism.

## 4. Discussion

Long-term HFD causes obesity in mice by disrupting glucose and lipid metabolism and altering gut microbiota, while dietary fibers and specific probiotics have demonstrated potential in mitigating these effects [26,27,28]. Fu brick tea, known for its unique fermentation process and health benefits, including blood glucose, lipid, and gut microbiota regulation. EC, the dominant fungal strain in FBT, contributes to its distinctive flavor and bioactivity. Although the anti-obesity effect of FBT has been established, the specific mechanisms by which EC modulates glucose-lipid metabolism and gut microbiota remain unclear, and its role as a primary anti-obesity component needs further study [29,30,31]. In the present study, eight weeks of EC supplementation markedly increased BAT mass and elevated UCP1 expression in HFD-fed mice, thereby augmenting thermogenic capacity and alleviating lipid metabolic disturbances. Collectively, these results indicate that EC may exert anti-obesity effects by enhancing energy expenditure and contributing to the maintenance of metabolic homeostasis. While our findings in this murine model provide valuable mechanistic insights into the anti-obesity effects of EC, several considerations must be addressed for human translation. There are inherent differences between mice and humans in physiology, digestive tract anatomy, and the composition and function of the gut microbiota, which could influence the response to EC intervention. Accumulating evidence has strongly implicated the gut microbiota in the pathogenesis of obesity [27,32].

This study investigated the role of EC in modulating the gut microbiota composition in HFD-fed obese mice. In obese mice, the *F*/*B* ratio in the gut microbiota decreased, with *Firmicutes* playing a key role in obesity-related metabolic disorders [33]. EC intervention reversed the *F*/*B* ratio (*p* < 0.05) in HFD mice, alleviating obesity symptoms. The abundance of *Proteobacteria* reflects gut microbiota instability, a hallmark of inflammation [34]. Compared to HFD mice, EC administration reduced the abundance of *Proteobacteria* (suggesting EC’s anti-inflammatory role via gut microbiota modulation) and significantly increased *Alloprevotella* (*p* < 0.05), indicating EC’s prebiotic effect. EC also notably decreased *Helicobacter pylori* levels (*p* < 0.05) elevated by HFD. Although *Helicobacter* has been reported to be associated with stomach problems in patients [35], further investigation is needed to determine whether it is related to obesity.

SCFAs, produced through gut microbial fermentation of dietary fibers, exert anti-obesity effects by directly or indirectly modulating lipid accumulation, inflammation, and insulin resistance [36,37]. EC supplementation boosted SCFA-producing bacteria and elevated fecal acetic acid, propionic acid, butyric acid, and succinate in HFD mice. Specifically, the high-dose EC group exhibited a 37.3% increase in total SCFAs compared to the HFD group (*p* < 0.01), implying that microbial metabolites contribute to the anti-obesity effects of EC. Studies have shown that IGN improves glucose and energy homeostasis, and IGN substrates (e.g., propionate, butyrate, succinate) can activate IGN to enhance glucose regulation [38,39]. Consistent with the increased abundance of beneficial bacteria and SCFAs production, EC significantly upregulated the expression of key IGN proteins (PEPCK, FBPase, and G6Pase) in the colon (*p* < 0.05). Untargeted metabolomics revealed that EC significantly regulated linoleic acid and other fatty acid metabolism pathways in HFD-fed mice. While these findings provide compelling hypotheses, future studies employing targeted quantitative assays (e.g., using LC-MS/MS with authentic standards) are warranted to absolutely confirm the identity and quantify the precise changes in these critical metabolites. These findings suggest that EC alleviates HFD-induced metabolic disorders by remodeling metabolic networks and attenuating lipid peroxidation and inflammatory responses [40].

This study reveals, for the first time, that EC markedly ameliorates glucose and lipid metabolic dysfunctions in HFD-induced obese mice by promoting the expansion of SCFA-producing gut microbial taxa, enhancing SCFA biosynthetic capacity, and activating the intestinal IGN signaling pathway. It is important to note that while our integrated multi-omics approach reveals strong associations between EC supplementation, specific shifts in the gut microbiota, increased production of SCFAs and succinate, and activation of IGN, the present study does not provide direct causal evidence. Future studies employing fecal microbiota transplantation from EC-treated mice into germ-free or antibiotic-treated recipients, or the monocolonization of gnotobiotic mice with key EC-enriched bacteria, will be crucial to definitively establish the causal role of the gut microbiota in mediating the metabolic benefits of EC. Addressing these issues will provide a critical foundation for developing precision nutrition interventions based on EC.

## 5. Conclusions

Our study demonstrates that supplementation with live EC significantly alleviates high-fat-diet-induced obesity and glucolipid metabolic disorders in mice by remodeling the gut microbiota, enhancing the production of short-chain fatty acids and succinate, and subsequently activating the intestinal gluconeogenesis pathway. These findings not only elucidate a novel mechanism behind the anti-obesity effects of EC but also position it as a promising probiotic candidate for functional food applications. Future research should focus on validating these results in human clinical trials, establishing causal relationships through fecal microbiota transplantation, and identifying the key bioactive metabolites responsible for the observed benefits. This work provides a scientific foundation for developing EC-based dietary strategies to combat metabolic diseases.

## Figures and Tables

**Figure 1 foods-14-04273-f001:**
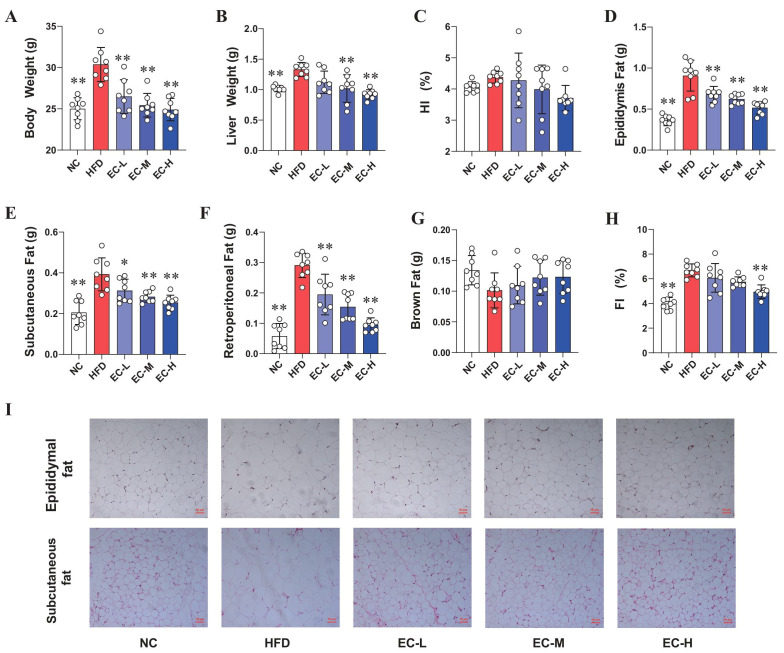
Effects of *Eurotium Cristatum* (EC) administration on the body weight, liver weight and fat weight in high-fat-diet (HFD) mice. (**A**) Body weight, (**B**) liver weight, (**C**) hepatic index (HI) = liver weight/body weight, (**D**) epididymis fat, (**E**) subcutaneous fat, (**F**) retroperitoneal fat, (**G**) brown fat, (**H**). Fat index (FI) = total fat weight/body weight. (**I**) The images of epididymis and subcutaneous adipose tissue H&E staining (original magnification of 200×). Data are presented as the mean ± standard deviation (SD). (*) *p* < 0.05 and (**) *p* < 0.01 vs. HFD group.

**Figure 2 foods-14-04273-f002:**
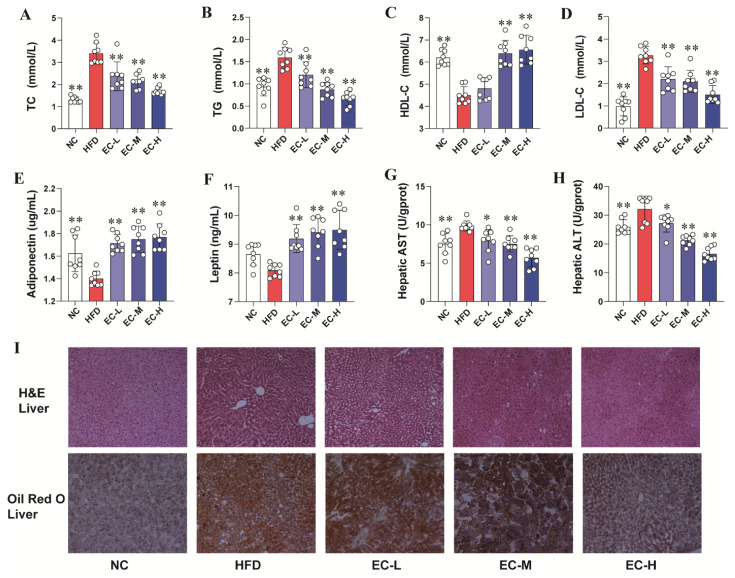
EC reduced serum dyslipidemia and improved liver injury in HFD-fed obese mice. (**A**) Serum TC levels, (**B**) serum TG levels, (**C**) serum HDL-C levels, (**D**) serum LDL-C levels, (**E**) serum adiponectin levels, (**F**) serum leptin levels, (**G**) hepatic AST, (**H**) hepatic ALT, (**I**) images of liver tissue H&E staining (original magnification of 200×) and Oil Red O staining (original magnification of 200×). Data are presented as the mean ± standard deviation (SD). (*) *p* < 0.05 and (**) *p* < 0.01 vs. HFD group.

**Figure 3 foods-14-04273-f003:**
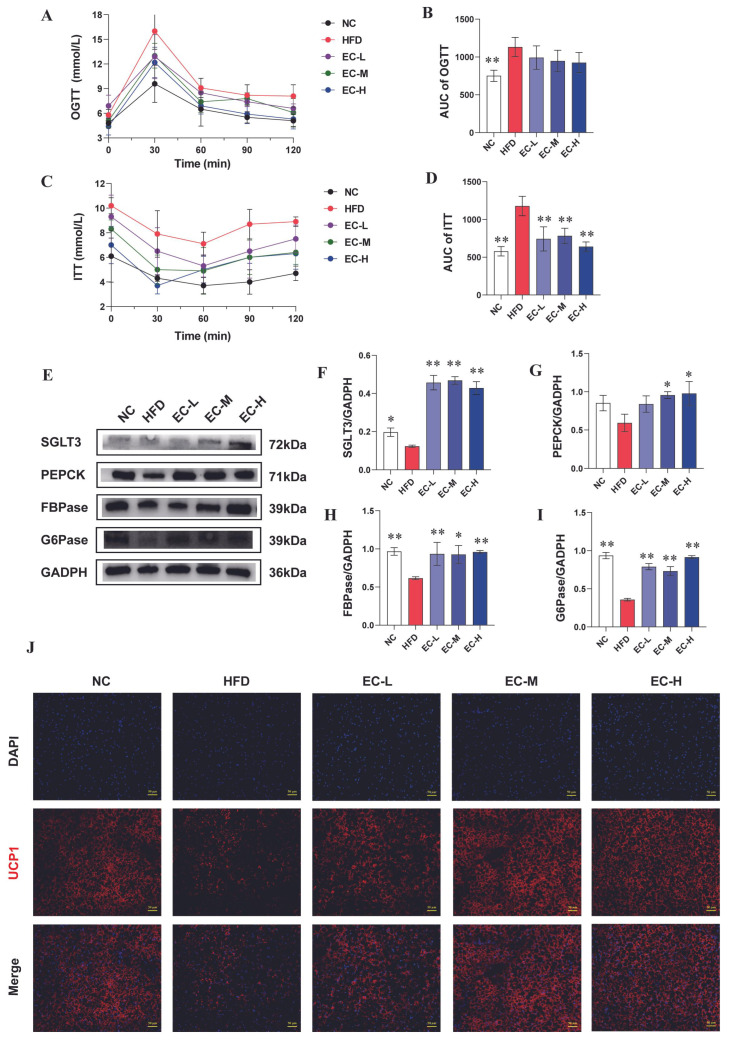
EC effectively improved glycolipid metabolism disorders in HFD mice. (**A**) OGTT, oral glucose tolerance test; (**B**) AUC of OGTT, (**C**) ITT, insulin sensitivity test; (**D**) AUC of ITT; (**E**–**I**) immunoblots of G6Pase, FBPase, PEPCK and SGLT3 in the colon; (**J**) regulatory effects of EC on protein expressions of UCP1 in brown fat that were investigated by using the immunofluorescence staining. Data are presented as the mean ± standard deviation (SD). (*) *p* < 0.05 and (**) *p* < 0.01 vs. HFD group.

**Figure 4 foods-14-04273-f004:**
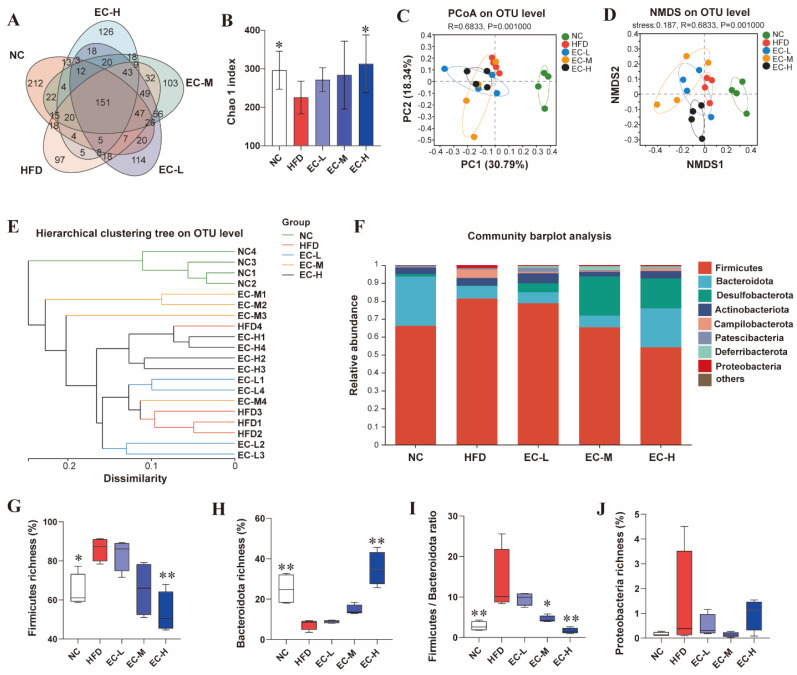
Modulatory effects of EC on the gut microbiota composition in HFD mice. (**A**) Venn diagram showing similarities between experimental groups based on OUT, (**B**) Chao 1 index, (**C**) PCoA of gut microbiota based on OUT relative abundance, (**D**) NMDS of gut microbiota based on OUT relative abundance, (**E**) hierarchical cluster analysis by the Euclidean distance matrix, (**F**) bacterial taxonomic profiling at the phylum level of gut microbiota, (**G**) relative abundance of *Firmicutes*, (**H**) relative abundance of *Bacteroidetes*, (**I**) the ratio of *Firmicutes*/*Bacteroidetes*, (**J**) relative abundance of *Proteobacteria*. Data are presented as the mean ± standard deviation (SD). (*) *p* < 0.05 and (**) *p* < 0.01 vs. HFD group.

**Figure 5 foods-14-04273-f005:**
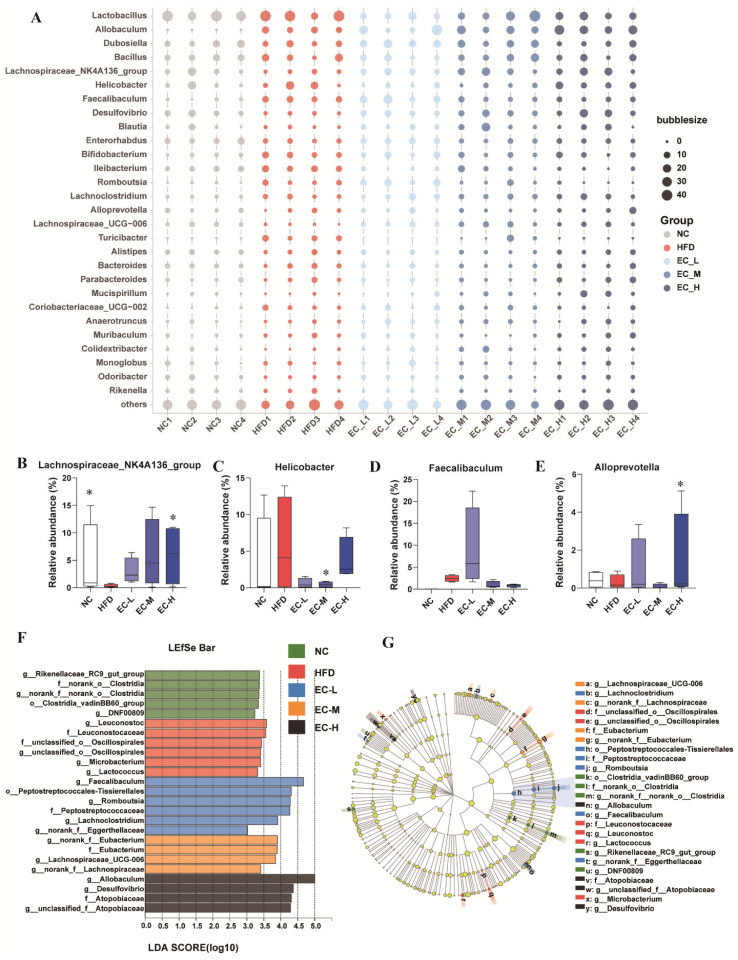
Differences in the abundances of the gut microbiota between the HFD-fed mice and the EC-treated mice. (**A**) Bubble chart showing relative abundances of gut microbiota at the genus level among experimental groups. The comparative analysis of the relative abundance of (**B**) *Lachnospiraece_NK4A136_group*, (**C**) *Helicobacter*, (**D**) *Faecalibaculum*, (**E**) *Alloprevotella*. (**F**,**G**) Linear discriminative analysis (LDA) effect size (LEfSe) analyses of statistically significant taxa. Taxa were sorted by the degree of difference and overlaid on a taxonomic cladogram. Only the taxa meeting a significant LDA threshold value of >3.0 are shown. Data are presented as the mean ± standard deviation (SD). (*) *p* < 0.05 vs. HFD group.

**Figure 6 foods-14-04273-f006:**
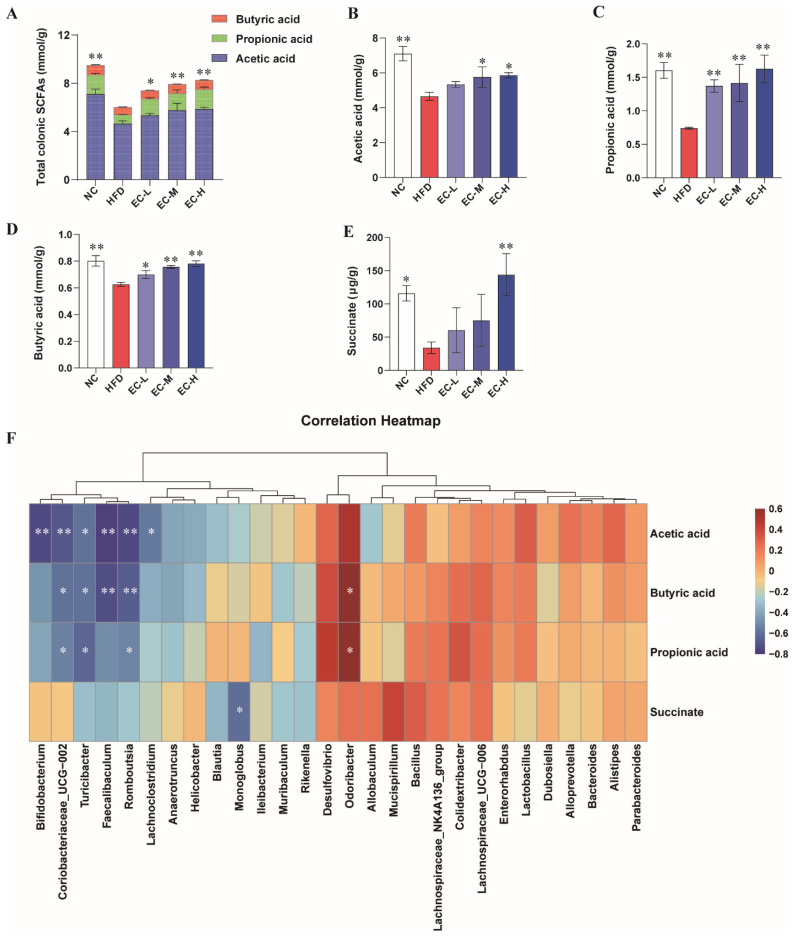
Effects of EC on SCFAs in the colon of HFD mice. (**A**) Total SCFAs levels, (**B**) acetic acid levels, (**C**) propionic acid levels, (**D**) butyric acid levels, (**E**) succinate levels. (**F**) Heatmap showing correlations of SCFAs and gut microbiota that were significantly affected by different treatments. Data are presented as the mean ± standard deviation (SD). (*) *p* < 0.05 and (**) *p* < 0.01 vs. HFD group.

**Figure 7 foods-14-04273-f007:**
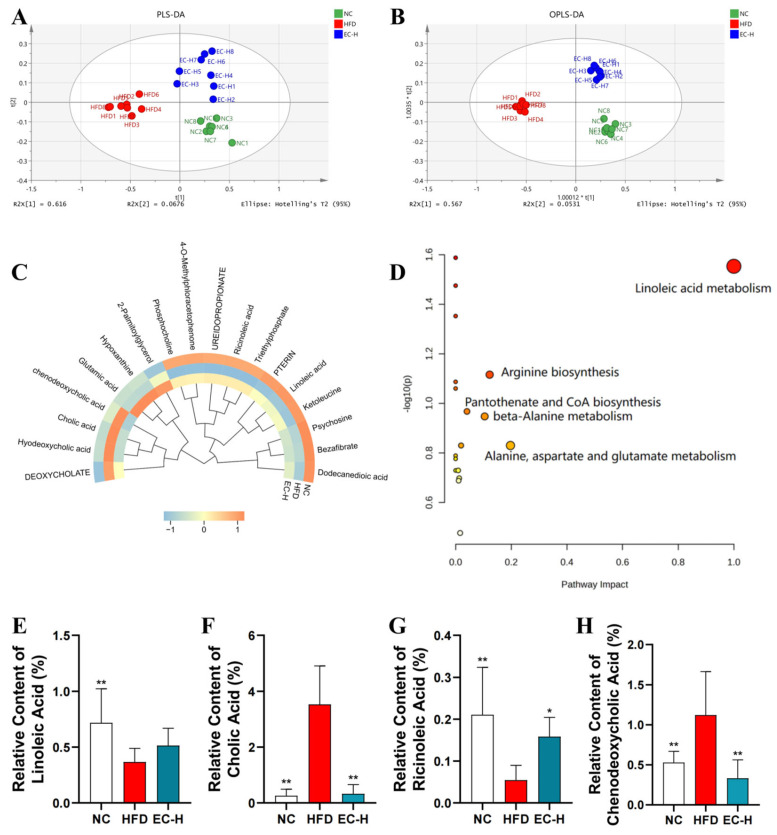
Effects of EC on colonic metabolites in obese mice. (**A**) PLS-DA. (**B**) OPLS-DA. (**C**) Circular Heatmap of differential metabolites among groups. (**D**) KEGG pathway enrichment analysis of differential metabolites. (**E**) Relative content of linoleic acid. (**F**) Relative content of cholic acid. (**G**) Relative content of ricinoleic acid. (**H**) Relative content of chenodeoxycholic acid. Data are presented as the mean ± standard deviation (SD). (*) *p* < 0.05 and (**) *p* < 0.01 vs. HFD group.

**Figure 8 foods-14-04273-f008:**
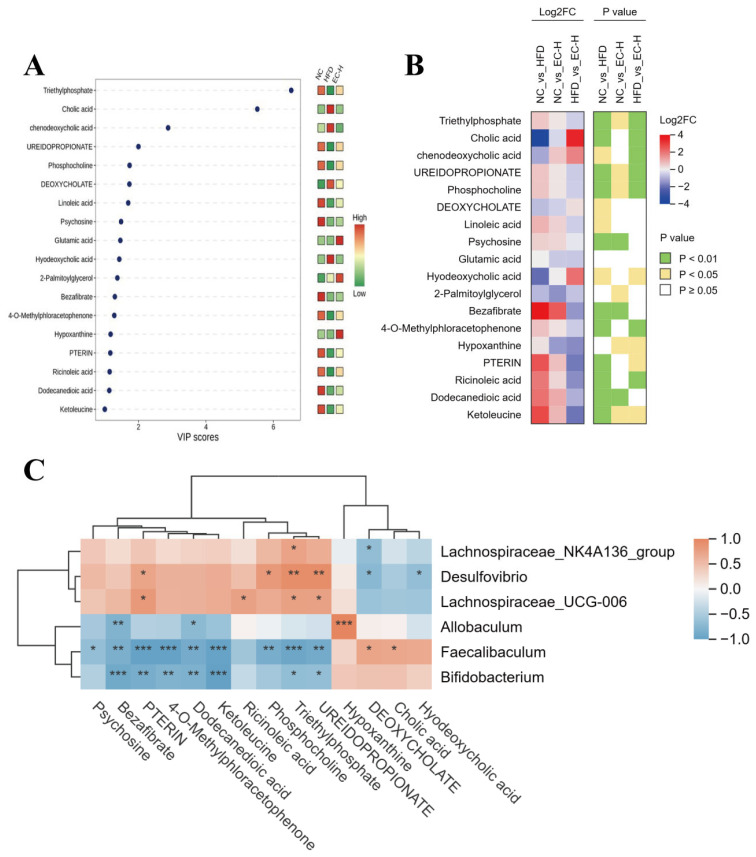
Effects of EC on colonic metabolites in obese mice. (**A**) Dot plot of differential metabolites with VIP values. (**B**) Combined heatmap of Log2FC values and *p* values. (**C**) Correlation analysis between the content of differential metabolites and the relative abundance of bacterial genera among different groups. Data are presented as the mean ± standard deviation (SD). (*) *p* < 0.05, (**) *p* < 0.01 and (***) *p* < 0.001 vs. HFD group.

## Data Availability

The original contributions presented in this study are included in the article/Appendix A. Further inquiries can be directed to the corresponding author.

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
