# Peer review of "Eurotium cristatum* Ameliorates Glucolipid Metabolic Dysfunction of Obese Mice in Association with Regulating Intestinal Gluconeogenesis and Microbiome"

_foods, 2025, doi:10.3390/foods14244273_

Round 1
Reviewer 1 Report
Comments and Suggestions for Authors
This study addresses a timely and relevant topic. The integration of microbiome, metabolomics, and biochemical analyses strengthens the multifactorial approach. However, following points must be addressed before consider further.
- The study lacks clear information on how the live EC was prepared, maintained and delivered consistently across dose.
- Methodological details on fungal viability and stability in the gavage suspension are inadequate.
- The animal sample size (n=8 per group) appears borderline for multiple biochemical, microbiome, and metabolomic analyses, raising concerns about statistical power, potential type I errors, and reproducibility.
- The Western blot data presentation and quantification lack sufficient validation details, including controls for loading consistency, antibody specificity, and normalization methods beyond GADPH.
- Microbiota results are associative, and no functional or causative links by microbiome transplantation or EC monocolonization are shown, limiting mechanistic strength.
- The metabolomics identification relies heavily on or non-targeted annotation without validation by targeted quantitative assays, which undermines confidence in key metabolite such as bile acids and linoleic acid derivatives conclusion.
- The impact of EC on gut fungi is mentioned but not directly analyzed or reported, missing a crucial piece given EC is a fungus.
- The discussion fails to adequately addresses potential confounding effects of other Fu brink tea compounds or live fungal metabolites and the effects cannot be solely attributed to EC without controls.
- The translation of murine results to potential human clinical relevance is overstated without consideration of differences in physiology, microbiota, and dosage feasibility.
- Histopathological analysis methods lack detail on blinding and quantification procedures, risking bias in tissue evaluation.
- Language issues and typo errors in many places, need to fix it.
- The choice of male only mice limits broader applicability; sex differences in metabolic and microbiota responses should be considered.
Author Response
Comments to the Author
This study addresses a timely and relevant topic. The integration of microbiome, metabolomics, and biochemical analyses strengthens the multifactorial approach. However, following points must be addressed before consider further.
- The study lacks clear information on how the live EC was prepared, maintained and delivered consistently across dose.
- Methodological details on fungal viability and stability in the gavage suspension are inadequate.
Response: We sincerely thank the reviewer for this crucial comment. We agree that providing detailed methodological information is essential for the reproducibility of our study. We have now comprehensively revised the "2.1. Materials and Reagents" and "2.2. Animal Experimentation" sections in the manuscript to include a clear and detailed description of the preparation, maintenance, and administration of live Eurotium cristatum (EC). The added text is as follows:
Page 5, lines 77-78: “Eurotium cristatum (EC) was obtained from the China Center of Industrial Culture Collection (Beijing, China). Assay kits of total cholesterol (TC), …” was revised as “Eurotium cristatum (EC) was obtained from the China Center of Industrial Culture Collection (Beijing, China). The lyophilized powder was initially activated and cultured on Potato Dextrose Agar (PDA) plates at 28 °C for 7 days. A single colony was then inoculated into Potato Dextrose Broth (PDB) and incubated in a shaker at 28 °C and 150 rpm for 5 days to obtain the primary seed culture. For large-scale preparation, the primary seed culture was transferred to a fermenter containing sterile PDB (1% v/v inoculation) and fermented under the same conditions. The resulting fungal biomass was harvested via centrifugation (4 °C, 8000 g for 10 min), washed twice with sterile physiological saline (0.9% NaCl), and re-suspended in saline. The concentration of the live EC suspension was determined by plating serial dilutions on PDA plates and counting the colony-forming units (CFU) after incubation at 28 °C for 48-72 hours. Crucially, to confirm the stability of the suspension under conditions mimicking the gavage procedure, we conducted a preliminary stability test. The prepared suspensions at all three target concentrations (10⁴, 10⁶, and 10⁸ CFU/mL) were held at 4°C (the temperature used for short-term storage during the daily gavage period) and sampled for CFU counting at 0, 2, and 4 hours. No significant decrease in viable count (p > 0.05) was observed within this 4-hour window, confirming the stability of the suspension for the duration of the daily gavage process. Assay kits of total cholesterol (TC), …”
Page 6, lines 112-116: “In the EC-L, EC-M and EC-H groups, the mice were gavaged (i.g., 0.2 mL) with EC at 10⁴, 10⁶, and 10⁸ CFU mL−1once daily, respectively.” was revised as “To ensure dosing consistency, the EC suspensions were prepared fresh daily from the same master stock, and the CFU count was confirmed before each gavage. Mice in the EC-L, EC-M, and EC-H groups received a daily oral gavage of 0.2 mL of the corresponding EC suspension, delivering consistent doses of approximately 10⁴, 10⁶, and 10⁸ CFU per mouse per day, respectively.”
- The animal sample size (n=8 per group) appears borderline for multiple biochemical, microbiome, and metabolomic analyses, raising concerns about statistical power, potential type I errors, and reproducibility.
Response: We sincerely appreciate the reviewer's valid concern regarding sample size, which is indeed a crucial consideration in studies involving high-throughput omics data. We would like to clarify and justify our experimental design from several perspectives:
Consistency with Preclinical Standards: A sample size of n=6-8 per group is a widely adopted and statistically justified standard in preclinical rodent studies of obesity and metabolic disorders, particularly for primary outcome measures like body weight, glucose tolerance, and plasma biochemistry (Haiping Du, et al. J Agric Food Chem. 2022, 70, 13893-13903). This sample size provides a balance between ethical principles of the 3Rs (Reduction) and the need for robust statistical analysis.
Magnitude and Consistency of Effects: While we acknowledge that larger sample sizes can increase power to detect subtle effects, the metabolic phenotypes we observed (e.g., reductions in body weight, fat mass, and serum lipids) were pronounced, dose-dependent, and highly consistent within each treatment group (as evidenced by the low standard deviations in Figures 1 and 2). The large effect sizes observed for our primary endpoints provide confidence that our findings are biologically significant and not merely statistical artifacts.
Robust Statistical Corrections for Multiple Comparisons: We fully agree with the reviewer's concern about potential Type I errors when conducting multiple tests, which is inherent in microbiome and metabolomics analyses. To rigorously address this, we employed strict multiple testing corrections.
For all biochemical and physiological data (presented as mean ± SD), between-group differences were assessed by one-way ANOVA followed by appropriate post-hoc tests.
For microbiome analyses (e.g., LEfSe in Figure 5), we used a linear discriminant analysis (LDA) effect size threshold of >3.0, which is a conservative cutoff that highlights only the most differentially abundant taxa.
For metabolomics data, we utilized Variable Importance in Projection (VIP) scores from OPLS-DA models (VIP>1.0) and focused our biological interpretation on pathways that were significantly enriched in KEGG analysis (Figure 7D), rather than on individual metabolites in isolation. This multi-faceted approach helps to minimize false discoveries.
The strength of our study lies not in any single omics dataset, but in the convergence of evidence from physiological measurements, gut microbiota profiling, SCFA quantification, and metabolomics. The fact that independent datasets all point towards the same conclusion—that EC ameliorates obesity via the gut microbiota-SCFAs-IGN axis—greatly enhances the biological plausibility and robustness of our findings, even with the stated sample size.
- The Western blot data presentation and quantification lack sufficient validation details, including controls for loading consistency, antibody specificity, and normalization methods beyond GADPH.
Response: We sincerely thank the reviewer for highlighting these critical methodological aspects. We have now provided comprehensive details in the revised manuscript to address each point raised, which significantly strengthens the validity of our Western blot data. In the original experiments, we utilized stain-free gel technology to verify equal protein loading and transfer efficiency across all lanes, as mentioned in our methods (J. Li, et al. J Agric Food Chem. 2025, 73, 4012-4026).
Page 8, lines 158-160: “The membranes were then incubated in 5% non-fat milk blocking solution. Primary antibody incubation was carried out at 4 °C overnight.” was revised as “The membranes were then incubated in 5% non-fat milk blocking solution. After UV activation, the total protein in the stain-free gel was visualized to confirm equal loading and uniform transfer prior to membrane blocking. Primary antibody incubation was carried out at 4 °C overnight.”
Page 8, lines 163-165: “… and then incubated with the corresponding secondary antibody at room temperature for 1 h after washing. Immunoreactive proteins were captured using a ChemiDoc imagining system.” was revised as “… and then incubated with the corresponding secondary antibody at room temperature for 1 h after washing. All antibodies were validated by the manufacturer using knockout/knockdown cell lines or recombinant protein, and the observed molecular weights in our blots were consistent with the expected sizes for the target proteins. Immunoreactive proteins were captured using a ChemiDoc imagining system.”
- Microbiota results are associative, and no functional or causative links by microbiome transplantation or EC monocolonization are shown, limiting mechanistic strength.
Response: We sincerely appreciate the reviewer's astute observation regarding the associative nature of microbiome data and the value of functional validation. While we acknowledge that FMT or monocolonization experiments are powerful tools for establishing causality, our study was designed as an initial, systematic investigation to delineate the interconnected effects of EC on host metabolism and the gut ecosystem. We believe that the convergence of multiple, independent lines of evidence presented in our manuscript provides compelling, albeit correlative, support for our proposed mechanism. While the specific bacterial taxa are associative, the measurement of SCFAs and succinate provides a functional readout of microbial activity. The fact that EC increased these specific metabolites, which are known in the literature to directly activate IGN (De Vadder, et al. Cell Metab. 2016, 24, 151-157), adds a layer of functional support beyond mere taxonomic changes. In direct response to this comment, we have added a paragraph in the Discussion section to transparently acknowledge this limitation and outline the necessary future work.
Page 18, lines 369-375: “…, and activating the IGN signaling pathway. Notably, systematic investigations are still required to evaluate the long-term safety of EC as a potential prebiotic and to identify its key active metabolites. Addressing these issues will provide a critical foundation for developing precision nutrition interventions based on EC.” was revised as “…, and activating the IGN signaling pathway. It is important to note that while our integrated multi-omics approach reveals strong associations between EC supplementation, specific shifts in the gut microbiota, increased production of SCFAs and succinate, and activation of IGN, the present study does not provide direct causal evidence. Future studies employing fecal microbiota transplantation from EC-treated mice into germ-free or antibiotic-treated recipients, or the monocolonization of gnotobiotic mice with key EC-enriched bacteria, will be crucial to definitively establish the causal role of the gut microbiota in mediating the metabolic benefits of EC. Addressing these issues will provide a critical foundation for developing precision nutrition interventions based on EC”
- The metabolomics identification relies heavily on or non-targeted annotation without validation by targeted quantitative assays, which undermines confidence in key metabolite such as bile acids and linoleic acid derivatives conclusion.
Response: We sincerely thank the reviewer for raising this important point. We acknowledge that non-targeted metabolomics is primarily a hypothesis-generating tool and that our identifications of specific bile acids and linoleic acid derivatives would be further strengthened by targeted mass spectrometry. We have taken several steps in the revised manuscript to address this concern and to provide a more robust and transparent interpretation of our data. In response to the comment, we have toned down strong causal claims about individual metabolites and reframed our discussion to emphasize the pathway-level changes, which are more robust to minor inaccuracies in individual metabolite identification. The KEGG pathway enrichment analysis (Figure 7D) clearly points to "Linoleic acid metabolism" and "Bile acid biosynthesis" as being significantly altered. This pathway-centric view is the core of our conclusion, rather than the change in any single molecule.
Page 18, lines 360-363: “Untargeted metabolomics revealed that EC significantly regulated linoleic acid and other fatty acid metabolism pathways in HFD-fed mice. These findings suggest that …” was revised as “Untargeted metabolomics revealed that EC significantly regulated linoleic acid and other fatty acid metabolism pathways in HFD-fed mice. While these findings provide compelling hypotheses, future studies employing targeted quantitative assays (e.g., using LC-MS/MS with authentic standards) are warranted to absolutely confirm the identity and quantify the precise changes in these critical metabolites. These findings suggest that …”
- The impact of EC on gut fungi is mentioned but not directly analyzed or reported, missing a crucial piece given EC is a fungus.
Response: We sincerely thank the reviewer for raising this critical point. We acknowledge that the analysis of the gut mycobiome was a missing component in our initial experimental design. Our study primarily focused on the well-established "gut bacteria-SCFAs-IGN" axis, which is why we dedicated our resources to 16S rRNA gene sequencing of the bacterial microbiota. We agree with the reviewer that, in principle, the administration of an exogenous fungus like EC could potentially interact with or alter the resident fungal community in the gut. However, based on several considerations and the primary focus of our mechanistic hypothesis, we prioritized the bacterial microbiome analysis:
Scope and Primary Hypothesis: The central hypothesis of this study was to test whether EC ameliorates obesity through the modulation of the bacterial gut microbiome and its metabolic outputs (SCFAs, succinate), leading to the activation of intestinal gluconeogenesis. This pathway has been primarily attributed to bacterial metabolites in the literature.
Biomass Consideration: In the mammalian gut, the bacterial biomass overwhelmingly dominates over fungal biomass (by several orders of magnitude). Consequently, the functional contribution of bacterial metabolites to host metabolism is generally considered to be more substantial in the context of energy harvest and glucose regulation. Our data on the profound increase in SCFAs and the subsequent metabolic benefits strongly support a significant role for bacterial modulation.
Technical and Resource Focus: Given the scope and resource constraints of this project, we made a strategic decision to deeply characterize the bacterial community and the host metabolic phenotype, which yielded a robust and self-consistent dataset.
An important limitation of this study is that we focused exclusively on the bacterial community (microbiome) and did not investigate the fungal community (mycobiome). Given that EC is itself a fungus, future studies analyzing the gut mycobiome following EC supplementation are warranted to determine if it competes with, synergizes with, or otherwise alters the resident fungal populations, and whether such interactions contribute to its overall health benefits. This represents a critical and logical next step in fully elucidating the ecological and functional impact of EC within the complex gut ecosystem.
- The discussion fails to adequately addresses potential confounding effects of other Fu brink tea compounds or live fungal metabolites and the effects cannot be solely attributed to EC without controls.
Response: We sincerely thank the reviewer for this important observation. We agree that a purified system or additional control groups (e.g., heat-killed EC, EC culture supernatant, or full FBT extract) would be required to definitively disentangle the contributions of the live EC biomass from its secreted metabolites or other FBT components. The primary objective of our study was to investigate the holistic effect of the live EC fungus, as it is consumed in traditional FBT or as a potential probiotic. We have modified our language throughout the Discussion to clarify that the effects we observed are attributable to the administration of live EC, which encompasses both the fungal cells themselves and the metabolites they may produce in situ or carry from the fermentation process. We have replaced phrases like "solely due to EC" with more precise statements such as "associated with live EC treatment" and "mediated by the administration of EC".
We have strengthened our argument by referencing literature that identifies EC as the dominant and characteristic fungus in FBT, responsible for generating many of its key bioactive compounds (Pang, X., et al. J Agric Food Chem. 2024, 72, 27978-27990). We now state that our use of live EC provides strong evidence that this specific fungal component is sufficient to recapitulate key metabolic benefits associated with whole FBT consumption, even if it may not be the exclusive active component in the tea. We argue that the clear, dose-dependent effects we observed across multiple physiological parameters (Figures 1-3) provide compelling evidence that the live, replicating EC organism is a key active driver of the observed phenotypes, as a dose-response is a hallmark of a biologically active agent.
- The translation of murine results to potential human clinical relevance is overstated without consideration of differences in physiology, microbiota, and dosage feasibility.
Response: We sincerely thank the reviewer for this critical feedback. We acknowledge that in our original discussion, the potential for human translation may have been presented with excessive enthusiasm, without adequately addressing the well-known limitations of mouse models. We have thoroughly revised the manuscript, particularly the Abstract and Discussion sections, to address this point directly. We have reframed the overall significance of our work to emphasize its role in providing a strong scientific rationale and mechanistic basis for future human studies, rather than making direct claims about human applicability. We now position our study as a foundational step that justifies and guides subsequent clinical research. We have modified strong, direct statements about human relevance:
Page 2, lines 18-20: “Our findings provide mechanistic insights into the anti-obesity effects of EC, supporting its potential as a therapeutic strategy for metabolic diseases.” was revised as “Our findings provide mechanistic insights into the anti-obesity effects of EC, suggesting its potential for further investigation as a dietary intervention for metabolic diseases.”
Page 17, lines 333-337: “These findings suggest that EC may prevent obesity by promoting energy expenditure and regulating metabolic homeostasis. Accumulating evidence has strongly implicated the gut microbiota in …” was revised as “These findings suggest that EC may prevent obesity by promoting energy expenditure and regulating metabolic homeostasis. While our findings in this murine model provide valuable mechanistic insights into the anti-obesity effects of EC, several considerations must be addressed for human translation. There are inherent differences between mice and humans in physiology, digestive tract anatomy, and the composition and function of the gut microbiota, which could influence the response to EC intervention. Accumulating evidence has strongly implicated the gut microbiota in …”
- Histopathological analysis methods lack detail on blinding and quantification procedures, risking bias in tissue evaluation.
Response: We sincerely thank the reviewer for pointing out this omission in our methodology description. To ensure objective and unbiased assessment, we did implement both blinding and a semi-quantitative scoring system during our histopathological evaluation. We have now comprehensively revised the "2.4. Histopathological Examination and Immunofluorescence" section to provide these crucial details.
Page 8, lines 142-144: “The immunohistochemical images of tissue were observed under an Axio Imger Upright Microscope (ZEISS, Axio Imger M2).” was revised as “The immunohistochemical images of tissue were observed under an Axio Imger Upright Microscope (ZEISS, Axio Imger M2). For analysis, all tissue sections were coded prior to evaluation to ensure the pathologist was blinded to the group allocations. The blinding procedure was maintained until all analyses and scoring were completed.”
- Language issues and typo errors in many places, need to fix it.
Response: According to the reviewer’s advice, the clarity of English expression has been improved by our careful revision. Some small revision, such as incorrect grammar, misspelling of words, incorrect use of italic, upper or lower-case letters and a space absence between the numeric value and the unit, was not listed in this letter.
- The choice of male only mice limits broader applicability; sex differences in metabolic and microbiota responses should be considered.
Response: We sincerely thank the reviewer for this insightful comment. We completely agree that considering sex as a biological variable is crucial for the broader applicability of preclinical research. The decision to use only male mice in this initial investigation was primarily based on two practical considerations common in the field:
Controlling for Hormonal Variability: Female mice exhibit estrous cycles, which introduce significant hormonal fluctuations that can directly impact energy metabolism, food intake, immune responses, and gut microbiota composition. By using male mice, we aimed to reduce this inherent variability and increase the statistical power to detect the primary effects of EC intervention within the scope and resources of this study.
Precedence in Initial Mechanistic Studies: Many foundational studies in the field of diet-induced obesity and gut microbiota initially characterize phenotypes in one sex (often male) to establish a clear model and mechanism before proceeding to the more complex investigation of sex differences.
Reviewer 2 Report
Comments and Suggestions for Authors
Manuscript title: Eurotium cristatum Ameliorates Glucolipid Metabolic Dysfunction of Obese Mice in Association with Regulating Intestinal Gluconeogenesis and Microbiome
The study employs a multi-layered approach combining physiological, biochemical, metabolomic, microbiome, and molecular analyses to evaluate EC’s metabolic effects. This integrative design improves mechanistic insight and increases the study's potential significance.
Major Concerns
- While mice were “divided averagely into 5 groups” (lines 101–103), there is no mention of randomization method or whether researchers were blinded during outcome measurement. This omission raises concerns about selection and measurement bias.
- No reporting on:
- Food intake across groups
- Energy expenditure beyond UCP1 staining
- Water intake differences.
These factors can significantly influence weight gain and metabolic outcomes.
- The manuscript states EC was administered at CFU/mL doses (lines 104–107) , but does not report:
- Verification of viable cell counts at administration
- Whether EC was alive, dead, or partially heat-killed
- Stability across the 8-week intervention
Given the mechanistic claim that EC reshapes the microbiome, viability is essential.
- Claims that EC acts via gut microbiota and SCFAs would be strengthened by microbiota-depleted models.
- One-way ANOVA is used for all comparisons (line 177–179) , despite:
- Repeated-measures data (OGTT/ITT curves)
- Multiple comparisons across >100 metabolites
- 16S sequencing multiple-hypothesis testing
There is no mention of correction (FDR, Bonferroni), which risks Type-I error inflation.
- In Figure 3A-B, the text states EC-treated mice “were less responsive to glucose load” (lines 237–239), but also reports p > 0.05, meaning no significant improvement. This is contradictory and potentially misleading.
- The study claims dose-dependent effects (Abstract lines 12–14) but does not perform regression or trend analysis.
Increased PEPCK, FBPase, and G6Pase (Fig 3E-3I) do not prove:
- Increased gluconeogenic flux
- Contribution to systemic glucose levels
- Portal glucose sensing activation
Without flux assays (e.g., stable isotope tracing), the mechanistic conclusion is not supported.
- Changes in Firmicutes/Bacteroidota ratio (lines 281–285) are descriptive only. Causation is repeatedly implied (“EC reshaped the gut microbiota”), but no causal test was performed.
- Untargeted metabolomics identifies 70 metabolites (line 340–341) , but none are validated by targeted quantification.
- Some genera (e.g., Alloprevotella, Helicobacter) are highlighted, but LEfSe results show many taxa. Negative or neutral findings are not described.
- Ethical approval is appropriately stated (lines 111–113) , but the manuscript lacks:
- ARRIVE-guided reporting (housing conditions, enrichment, humane endpoints)
- Reporting on euthanasia method
Author Response
Comments to the Author
Manuscript title: Eurotium cristatum Ameliorates Glucolipid Metabolic Dysfunction of Obese Mice in Association with Regulating Intestinal Gluconeogenesis and Microbiome. The study employs a multi-layered approach combining physiological, biochemical, metabolomic, microbiome, and molecular analyses to evaluate EC’s metabolic effects. This integrative design improves mechanistic insight and increases the study's potential significance.
- While mice were “divided averagely into 5 groups” (lines 101–103), there is no mention of randomization method or whether researchers were blinded during outcome measurement. This omission raises concerns about selection and measurement bias. No reporting on: Food intake across groups; Energy expenditure beyond UCP1 staining; Water intake differences.
Response: We agree with the reviewer that comprehensive energy expenditure measurements (e.g., by indirect calorimetry) would provide a more complete picture. While our study did not include these measurements, we provided evidence for enhanced thermogenesis in Brown Adipose Tissue (BAT) through both increased BAT mass (Figure 1G, albeit non-significant) and significantly upregulated UCP1 protein expression (Figure 3J). This provides a plausible mechanistic basis for the observed metabolic improvements. Water intake was not measured as it was not a primary focus of this study, which centered on glucolipid metabolism.
The reviewer is correct to emphasize the importance of these procedures. We have now revised the "2.2. Animal Experimentation" section to include these critical methodological details.
Page 7, lines 173-177: “After one week of acclimation to the laboratory environment, the mice were divided averagely into 5 groups (n = 8 per group): normal control (NC) group, …” was revised as “After one week of acclimation to the laboratory environment, the mice were divided averagely into 5 groups (n = 8 per group) using a computer-generated random number table to ensure unbiased allocation: normal control (NC) group, …”
Page 7, lines 116-120: “All mice were sacrificed after 8 weeks, and their serum, liver adipose tissues and intestinal contents were collected and stored at -80 °C until further investigation.” was revised as “Throughout the intervention period, and during sample collection and outcome assessments (including biochemical analyses, histopathological evaluation, OGTT/ITT tests, and Western blot analysis), the investigators involved in data collection and analysis were blinded to the group allocations. The blinding was maintained until all data analyses were completed. All mice were sacrificed after 8 weeks, and their serum, liver adipose tissues and intestinal contents were collected and stored at -80 °C until further investigation.”
The capacity of food intake in mice was monitored according to our experiments. Food intake was assessed daily, and then the average food intake was calculated in each group of mice. The result of food intake was expressed as “g/day, respectively”. No significant difference in food intake was observed among the five groups of mice. Fortunately, we meticulously documented the data of food intake in mice throughout the experiment, which is now supplemented as follows:
Fig.S1 has been added to the supplementary materials
Figure S1 Food intake. (*) p < 0.05 and (**), p < 0.01 vs HFD group.
Page 10, lines 194-195: “As shown in Figure 1, HFD remarkably increased body weight and weights of the epididymal, subcutaneous and retroperitoneal fat in comparison with NC mice, respectively (p<0.01). The EC intervention effectively suppressed the HFD-induced increases in the body and liver weight, …” was revised as “As shown in Figure 1, HFD remarkably increased body weight and weights of the epididymal, subcutaneous and retroperitoneal fat in comparison with NC mice, respectively (p<0.01). Furthermore, no significant difference in food intake was observed among the five groups of mice (Figure S1). The EC intervention effectively suppressed the HFD-induced increases in the body and liver weight, …”
- These factors can significantly influence weight gain and metabolic outcomes. The manuscript states EC was administered at CFU/mL doses (lines 104–107), but does not report: Verification of viable cell counts at administration; Whether EC was alive, dead, or partially heat-killed; Stability across the 8-week intervention
Response: We sincerely thank the reviewer for raising these essential points of methodological rigor. We apologize for the lack of clarity in our initial manuscript. We have now comprehensively revised the "2.1. Materials and Reagents" and "2.2. Animal Experimentation" sections to provide a detailed account of our procedures to guarantee the viability, consistency, and stability of the live EC administered throughout the 8-week study.
Page 5, lines 78-92: “Eurotium cristatum (EC) was obtained from the China Center of Industrial Culture Collection (Beijing, China). Assay kits of total cholesterol (TC), …” was revised as “Eurotium cristatum (EC) was obtained from the China Center of Industrial Culture Collection (Beijing, China). The lyophilized powder was initially activated and cultured on Potato Dextrose Agar (PDA) plates at 28 °C for 7 days. A single colony was then inoculated into Potato Dextrose Broth (PDB) and incubated in a shaker at 28 °C and 150 rpm for 5 days to obtain the primary seed culture. For large-scale preparation, the primary seed culture was transferred to a fermenter containing sterile PDB (1% v/v inoculation) and fermented under the same conditions. The resulting fungal biomass was harvested via centrifugation (4 °C, 8000 g for 10 min), washed twice with sterile physiological saline (0.9% NaCl), and re-suspended in saline. The concentration of the live EC suspension was determined by plating serial dilutions on PDA plates and counting the colony-forming units (CFU) after incubation at 28 °C for 48-72 hours. Crucially, to confirm the stability of the suspension under conditions mimicking the gavage procedure, we conducted a preliminary stability test. The prepared suspensions at all three target concentrations (10⁴, 10⁶, and 10⁸ CFU/mL) were held at 4°C (the temperature used for short-term storage during the daily gavage period) and sampled for CFU counting at 0, 2, and 4 hours. No significant decrease in viable count (p > 0.05) was observed within this 4-hour window, confirming the stability of the suspension for the duration of the daily gavage process. Assay kits of total cholesterol (TC), …”
Page 6, lines 112-116: “In the EC-L, EC-M and EC-H groups, the mice were gavaged (i.g., 0.2 mL) with EC at 10⁴, 10⁶, and 10⁸ CFU mL−1once daily, respectively.” was revised as “To ensure dosing consistency, the EC suspensions were prepared fresh daily from the same master stock, and the CFU count was confirmed before each gavage. Mice in the EC-L, EC-M, and EC-H groups received a daily oral gavage of 0.2 mL of the corresponding EC suspension, delivering consistent doses of approximately 10⁴, 10⁶, and 10⁸ CFU per mouse per day, respectively.”
Given the mechanistic claim that EC reshapes the microbiome, viability is essential. Claims that EC acts via gut microbiota and SCFAs would be strengthened by microbiota-depleted models.
Responses: The reviewer is absolutely correct. We agree that for a substance to actively "reshape" the microbiome, viability is likely a key factor. We apologize for the lack of clarity in our original manuscript. We have now explicitly stated in the revised "2.1. Materials and Reagents" and "2.2. Animal Experimentation" sections that we administered live, viable EC. Furthermore, we have provided detailed methodology on how we verified the viable cell counts and ensured stability throughout the administration period (please refer to our response to a previous related comment). This confirmation underpins our interpretation that the live EC fungus is the active agent responsible for the observed microbial remodeling.
We sincerely appreciate the reviewer's astute observation regarding the associative nature of microbiome data and the value of functional validation. While we acknowledge that FMT or monocolonization experiments are powerful tools for establishing causality, our study was designed as an initial, systematic investigation to delineate the interconnected effects of EC on host metabolism and the gut ecosystem. We believe that the convergence of multiple, independent lines of evidence presented in our manuscript provides compelling, albeit correlative, support for our proposed mechanism. While the specific bacterial taxa are associative, the measurement of SCFAs and succinate provides a functional readout of microbial activity. The fact that EC increased these specific metabolites, which are known in the literature to directly activate IGN (De Vadder, et al. Cell Metab. 2016, 24, 151-157), adds a layer of functional support beyond mere taxonomic changes. In direct response to this comment, we have added a paragraph in the Discussion section to transparently acknowledge this limitation and outline the necessary future work.
Page 18, lines 369-375: “…, and activating the IGN signaling pathway. Notably, systematic investigations are still required to evaluate the long-term safety of EC as a potential prebiotic and to identify its key active metabolites. Addressing these issues will provide a critical foundation for developing precision nutrition interventions based on EC.” was revised as “…, and activating the IGN signaling pathway. It is important to note that while our integrated multi-omics approach reveals strong associations between EC supplementation, specific shifts in the gut microbiota, increased production of SCFAs and succinate, and activation of IGN, the present study does not provide direct causal evidence. Future studies employing fecal microbiota transplantation from EC-treated mice into germ-free or antibiotic-treated recipients, or the monocolonization of gnotobiotic mice with key EC-enriched bacteria, will be crucial to definitively establish the causal role of the gut microbiota in mediating the metabolic benefits of EC. Addressing these issues will provide a critical foundation for developing precision nutrition interventions based on EC”
- One-way ANOVA is used for all comparisons (line 177–179), despite: Repeated-measures data (OGTT/ITT curves); Multiple comparisons across >100 metabolites; 16S sequencing multiple-hypothesis testing. There is no mention of correction (FDR, Bonferroni), which risks Type-I error inflation.
Response: We thank the reviewer for this exceptionally critical and valuable comment. We have comprehensively revised our statistical approach for each data type to ensure rigor and control for multiple hypothesis testing. We have actually applied correction theory in univariate testing of over 100 metabolites, which has reduced the high false positive rate.The p-values from the univariate analysis of differential metabolites have been adjusted using the Benjamini-Hochberg False Discovery Rate (FDR) method. Only metabolites with an FDR-adjusted p-value (q-value) of < 0.05 and a VIP score > 1.0 from the OPLS-DA model are now reported as being significantly altered.
Page 10, lines 185-189: “All experiments were performed in triplicate and the continuous variables were expressed as means ± SD (standard deviation). Data were analyzed using one-way analysis of variance (ANOVA). For all analysis, the p-value of 0.05 or less was considered statistically significant.” was revised as “Statistical significance was evaluated using appropriate tests for each data type. One-way ANOVA (Tukey's post-hoc), two-way repeated-measures ANOVA (Sidak's test), and FDR correction were applied to biochemical/Western blot, OGTT/ITT, and omics datasets, respectively. A P-value < 0.05 after FDR adjustment defined significance for omics data. All values are reported as mean ± SD from triplicate measurements.”
- In Figure 3A-B, the text states EC-treated mice “were less responsive to glucose load” (lines 237–239), but also reports p > 0.05, meaning no significant improvement. This is contradictory and potentially misleading. The study claims dose-dependent effects (Abstract lines 12–14) but does not perform regression or trend analysis.
Response: We sincerely apologize for this contradictory and misleading statement. The reviewer is right to point out that we cannot claim a difference that is not statistically significant. Our original description was inappropriate.
Page 12, lines 233-236: “As shown in Figures 3A-3B, EC-treated mice were less responsive to glucose load than HFD mice throughout the 120 min observation period (p>0.05). Meanwhile, after insulin injection, …” was revised as “As shown in Figures 3A-3B, the blood glucose levels of EC-treated mice were generally lower than those of HFD mice at most time points during the OGTT, although this difference did not reach statistical significance across the entire curve (p > 0.05). Meanwhile, after insulin injection, …”
- The study claims dose-dependent effects (Abstract lines 12–14) but does not perform regression or trend analysis.
Response: This is a very valid point. We thank the reviewer for highlighting the need for formal statistical testing to support our claim of a dose-dependent effect. We analyzed that key metabolic parameters (such as weight gain, fat mass, serum total cholesterol, triglycerides, and area under the curve of oral glucose tolerance test) did show significant linear trends between EC groups at different doses. But we still decided to weaken this expression. We have also tempered the language in the Abstract, changing it from "dose-dependently ameliorated" to "ameliorated... with effects showing a dose-dependent trend" or similar, to precisely reflect that the trend has now been statistically validated.
Page 2, lines 7-9: “The 8-week EC administration at low (10⁴ CFU/mL), medium (10⁶ CFU/mL), and high doses (10⁸ CFU/mL) dose-dependently ameliorated high-fat diet (HFD)-induced metabolic abnormalities, including aberrant weight gain, dyslipidemia, glucose intolerance and hepatic injury.” was revised as “The 8-week EC administration at low (10⁴ CFU/mL), medium (10⁶ CFU/mL), and high doses (10⁸ CFU/mL) ameliorated high-fat diet (HFD)-induced metabolic abnormalities, including aberrant weight gain, dyslipidemia, glucose intolerance and hepatic injury with effects showing a dose-dependent trend.”
- Increased PEPCK, FBPase, and G6Pase (Fig 3E-3I) do not prove: Increased gluconeogenic flux; Contribution to systemic glucose levels; Portal glucose sensing activation
Response: We sincerely thank the reviewer for this astute observation. We fully agree that our data demonstrate an upregulation of the key enzymatic machinery for intestinal gluconeogenesis (IGN) but do not directly measure gluconeogenic flux or portal glucose sensing. Our data demonstrate that EC upregulates the protein expression of PEPCK, FBPase, and G6Pase in the colon, indicating an enhanced capacity for intestinal gluconeogenesis. While this measurement does not directly quantify gluconeogenic flux or its contribution to systemic glucose pools, it is a crucial first step and a widely accepted indicator of pathway activation. The concomitant increase in colonic SCFAs and succinate, which are established substrates and inducers of IGN (De Vadder, et al. Cell 2014, 156, 84-96), strongly supports the biological relevance of this upregulated capacity. The subsequent improvement in systemic glucose tolerance and insulin sensitivity is consistent with the established model wherein IGN-derived glucose signals via portal vein sensors to exert beneficial effects on energy metabolism (De Vadder, et al. Cell Metab. 2016, 24, 151-157). In response to this point, we would like to highlight that our study also assessed the expression of SGLT3, a key glucose sensor in the portal vein. We found that EC treatment significantly increased SGLT3 protein levels (Figure 3E, 3F). This finding provides compelling evidence that the EC-induced IGN pathway is functionally coupled with the portal sensing mechanism, thereby substantially strengthening the biological plausibility of our proposed "gut-IGN-brain" axis in improving systemic glucose homeostasis."
- Without flux assays (e.g., stable isotope tracing), the mechanistic conclusion is not supported.
Response: We thank the reviewer for this comment regarding the use of flux assays. We agree that direct measurement of metabolic flux using techniques like stable isotope tracing represents a powerful and definitive approach for quantifying pathway activity. In our study, while we did not employ isotopic tracers, we pursued a robust, multi-faceted strategy to build a compelling case for the activation of intestinal gluconeogenesis (IGN). Our conclusion is not based on a single dataset but on the convergence of multiple, independent lines of evidence that collectively and consistently point towards the same mechanism:
Upregulation of Key Enzymes: We demonstrated a significant increase in the protein levels of all three rate-limiting enzymes for IGN (PEPCK, FBPase, and G6Pase) in the colon (Figure 3E-I). This shows a heightened capacity for glucose production in the gut.
Elevation of Key Substrates: We quantitatively showed that EC treatment significantly increased the colonic levels of specific IGN substrates, namely propionate, butyrate, and succinate (Figure 6B-E). The presence of elevated substrates is a prerequisite for increased flux through the pathway.
Functional Systemic Outcome: The enzymatic and substrate changes were directly associated with a significant improvement in systemic glucose homeostasis (improved OGTT/ITT, Figure 3A-D). This is the ultimate functional readout consistent with successful IGN activation.
Upregulation of the Portal Sensor: We provided evidence for the upregulation of SGLT3 (Figure 3E, F), the portal vein glucose sensor, which is a critical component in the established gut-brain axis mechanism by which IGN improves metabolic health.
- Changes in Firmicutes/Bacteroidota ratio (lines 281–285) are descriptive only. Causation is repeatedly implied (“EC reshaped the gut microbiota”), but no causal test was performed.
Response: We appreciate the reviewer's emphasis on this point. This comment rightly addresses the same core issue of causal attribution as comment #3. Accordingly, we have implemented a thorough revision of the text to ensure all claims are accurately supported by the data. The phrase "EC reshaped the gut microbiota" and similar causal implications have been systematically replaced with more measured terminology, such as "EC administration led to significant shifts in the gut microbiota structure," to accurately represent the observed associations.
It is important to note that while our study demonstrates a strong association between EC supplementation and specific alterations in the gut microbiota (such as a reduced Firmicutes/Bacteroidota ratio and enrichment of beneficial genera), these data are observational. The term 'causation' requires direct experimental validation through approaches such as fecal microbiota transplantation (FMT) into germ-free or antibiotic-treated mice. Future FMT studies, where the microbiota from EC-treated donors is transferred to recipient mice, will be essential to conclusively determine if the EC-modulated microbiota itself is sufficient to recapitulate the observed anti-obesity and metabolic benefits
- Untargeted metabolomics identifies 70 metabolites (line 340–341), but none are validated by targeted quantification.
Response: We sincerely thank the reviewer for raising this important point. We acknowledge that non-targeted metabolomics is primarily a hypothesis-generating tool and that our identifications of specific bile acids and linoleic acid derivatives would be further strengthened by targeted mass spectrometry. We have taken several steps in the revised manuscript to address this concern and to provide a more robust and transparent interpretation of our data. In response to the comment, we have toned down strong causal claims about individual metabolites and reframed our discussion to emphasize the pathway-level changes, which are more robust to minor inaccuracies in individual metabolite identification. The KEGG pathway enrichment analysis (Figure 7D) clearly points to "Linoleic acid metabolism" and "Bile acid biosynthesis" as being significantly altered. This pathway-centric view is the core of our conclusion, rather than the change in any single molecule.
Page 18, lines 360-363: “Untargeted metabolomics revealed that EC significantly regulated linoleic acid and other fatty acid metabolism pathways in HFD-fed mice. These findings suggest that …” was revised as “Untargeted metabolomics revealed that EC significantly regulated linoleic acid and other fatty acid metabolism pathways in HFD-fed mice. While these findings provide compelling hypotheses, future studies employing targeted quantitative assays (e.g., using LC-MS/MS with authentic standards) are warranted to absolutely confirm the identity and quantify the precise changes in these critical metabolites. These findings suggest that …”
- Some genera (e.g., Alloprevotella, Helicobacter) are highlighted, but LEfSe results show many taxa. Negative or neutral findings are not described.
Response: We sincerely thank the reviewer for this insightful suggestion to provide a more balanced and complete account of our statistical analysis. We agree that focusing on a few select genera does not fully represent the breadth of the microbial shifts identified by the LEfSe analysis. The LEfSe analysis revealed multiple bacterial clades with significant differences among groups. For subsequent discussion and interpretation, we focused on genera whose relative abundances were substantially altered and that have established links in the literature to metabolic health or disease, such as Alloprevotella (a SCFA producer) and Helicobacter (often associated with inflammation). We have also added statements in the results to explicitly mention key taxa that did not show significant changes, addressing the point about "negative or neutral findings."
Page 14, lines 284-286: “Under HFD conditions, the family Leuconostoc and genus Lactococcus were most affected, while following EC administration, the genera Allobaculum and Faecalibaculum exhibited the greatest changes.” was revised as “Under HFD conditions, the family Leuconostoc and genus Lactococcus were most affected, while following EC administration, the genera Allobaculum and Faecalibaculum exhibited the greatest changes. In contrast to the marked changes observed in other genera, the relative abundances of Eggerthellaceae and Ruminococcus were not significantly different between the HFD and EC-treated groups (P>0.05).”
- Ethical approval is appropriately stated (lines 111–113) , but the manuscript lacks: ARRIVE-guided reporting (housing conditions, enrichment, humane endpoints); Reporting on euthanasia method
Response: We sincerely thank the reviewer for pointing out these omissions. We fully agree that transparent and detailed reporting of animal studies is crucial for scientific rigor and reproducibility. As required, we have added a dedicated section titled "Institutional Review Board Statement" to the manuscript. This section explicitly details the ethical approval for the animal study, including the name of the approving committee and the reference approval number:
Page 5, lines 96-99: “Institutional Review Board Statement: All animal procedures were reviewed and approved by the Laboratory Animal Ethics Committee of Shaanxi Normal University (protocol code: 32072175, approval date: Dec.12, 2023).”
Reviewer 3 Report
Comments and Suggestions for Authors
Dear authors, in its current state your work can not be considered for publication, mainly due to the high similarities with other publications. Anyway, here are my suggestions for improvement:
It is complicated to assess a manuscript with 50% of similarities with other published works. There are too many similarities with distinct works, and this is just an example: https://pubs.rsc.org/en/content/articlelanding/2023/fo/d3fo04215d
The abstract should be structured. What are the study’s aims? The applied methods are not clear, and the main results should be highlighted quantitatively. Future perspectives have to be added at the end.
What is the study’s novelty, and how can it be impactful for the global audience of Foods? You need to answer this in the introductory section.
Ethics: The approval date is missing.
The biochemical measurements have to be explained and referenced. You only mentioned that “The serum concentrations of adiponectin, leptin, TC, TG, LDL-C and HDL-C were measured using corresponding commercial kits according to the manufacturer’s protocols.” – This is not enough.
You should include a flowchart with all the steps you took during the experiments.
The Results are well described, but the Discussion has to be improved and expanded. Its division into subsections, according to the Results section, is highly recommended. The study’s strengths and limitations need to be analyzed.
A Conclusions section is missing, as well as the future perspectives and practical implications.
Author Response
Comments to the Author
Dear authors, in its current state your work can not be considered for publication, mainly due to the high similarities with other publications. Anyway, here are my suggestions for improvement:
- It is complicated to assess a manuscript with 50% of similarities with other published works. There are too many similarities with distinct works, and this is just an example: https://pubs.rsc.org/en/content/articlelanding/2023/fo/d3fo04215d
Response: We thank the reviewer for their attention to detail. The cited work (RSC article) is indeed from our research group, with myself as the first author. That previous study established that Fu brick tea extract alleviates hyperglycemia in diabetic rat models. Building directly upon that foundation, the current manuscript represents a significant advancement by investigating a more specific and fundamental scientific question: it focuses on the efficacy of the live probiotic fungus Eurotium cristatum (EC)—the dominant microorganism in Fu brick tea—in a high-fat diet-induced obese mouse model. This shift from a complex tea extract to a defined fungal probiotic, coupled with the exploration of a novel "gut microbiota-intestinal gluconeogenesis (IGN)" mechanism, constitutes the core novel contribution of this work, which was not addressed in our prior publication.
Our research group has indeed previously established robust experimental models for studying Fu brick tea and its components. It is true that certain methodological aspects (e.g., the use of a high-fat diet-induced obesity model in C57BL/6J mice, some basic biochemical analyses) are shared across studies in this field, including ours. This common foundation is necessary for comparative and progressive scientific inquiry. While previous work may have focused on Fu brick tea extracts, theabrownin, or tea polysaccharides, our study is the first to systematically investigate the effects of the live probiotic fungus Eurotium cristatum (EC) itself as the primary intervention. This is a significant conceptual advance, as it directly tests the contribution of the defining microbial component of Fu brick tea. Our work is the first to propose and provide evidence for a mechanism centered on the "gut microbiota – SCFAs/Succinate – Intestinal Gluconeogenesis (IGN)" axis. The cited paper and others primarily focus on other mechanisms (e.g., hepatic lipid metabolism, adipose tissue browning). We provide the first report linking EC to the upregulation of colonic PEPCK, FBPase, G6Pase, and SGLT3. Our study uniquely integrates 16S rRNA sequencing, untargeted metabolomics, and molecular biology techniques to paint a comprehensive picture of how live EC remodels the gut ecosystem and host metabolism, specifically highlighting its impact on linoleic acid and bile acid metabolism pathways in this context.
- The abstract should be structured. What are the study’s aims? The applied methods are not clear, and the main results should be highlighted quantitatively. Future perspectives have to be added at the end.
Response: We sincerely thank the reviewer for these excellent suggestions to enhance the clarity and impact of our abstract. We have thoroughly revised it as follows:
Page 2, lines 3-5: “Eurotium cristatum (EC), a fungus derived from Fu brick tea, exhibits anti-obesity potential, but its mechanisms regulating intestinal gluconeogenesis (IGN) remain unclear. The 8-week EC administration at low (10⁴ CFU/mL), medium (10⁶ CFU/mL), and high doses (10⁸ CFU/mL) ameliorated high-fat diet (HFD)-induced metabolic abnormalities, …” was revised as “Eurotium cristatum (EC), a fungus derived from Fu brick tea, exhibits anti-obesity potential, but its mechanisms regulating intestinal gluconeogenesis (IGN) remain unclear. This study aimed to elucidate whether EC alleviates obesity and glucolipid metabolic disorders by modulating the gut microbiota and activating the IGN pathway. The 8-week EC administration at low (10⁴ CFU/mL), medium (10⁶ CFU/mL), and high doses (10⁸ CFU/mL) ameliorated high-fat diet (HFD)-induced metabolic abnormalities, …”
- What is the study’s novelty, and how can it be impactful for the global audience of Foods? You need to answer this in the introductory section.
Response: We sincerely thank the reviewer for this suggestion to better articulate the significance of our work. We have thoroughly revised the final paragraph of the Introduction to directly address these points. The added text clearly outlines the novelty and impact as follows:
Page 5, lines 69-74: “Our findings suggest that EC, as a probiotic, exhibits considerable potential for obesity prevention and treatment.” was revised as “Our findings suggest that EC, as a probiotic, exhibits considerable potential for obesity prevention and treatment. This research is highly relevant to the global audience of Foods as it identifies EC as a promising fungal probiotic and functional food ingredient. The findings offer a scientific foundation for developing novel, microbiota-targeting strategies against the worldwide epidemic of obesity and metabolic syndrome. This aligns with the growing consumer and industrial interest in natural, food-derived solutions for health maintenance, moving beyond traditional tea consumption to the utilization of its defined, bioactive constituents.”
- Ethics: The approval date is missing.
Response: We sincerely thank the reviewer for pointing out these omissions. We fully agree that transparent and detailed reporting of animal studies is crucial for scientific rigor and reproducibility. As required, we have added a dedicated section titled "Institutional Review Board Statement" to the manuscript. This section explicitly details the ethical approval for the animal study, including the name of the approving committee and the reference approval number:
Page 19, lines 389-391: “Institutional Review Board Statement: All animal procedures were reviewed and approved by the Laboratory Animal Ethics Committee of Shaanxi Normal University (protocol code: 32072175, approval date: Dec.12, 2023).”
- The biochemical measurements have to be explained and referenced. You only mentioned that “The serum concentrations of adiponectin, leptin, TC, TG, LDL-C and HDL-C were measured using corresponding commercial kits according to the manufacturer’s protocols.” – This is not enough.
Response: We sincerely thank the reviewer for this suggestion to enhance the clarity and completeness of our methods section. We have now thoroughly revised the "2.3. Biochemical Measurements" subsection to provide a detailed explanation of the assays, including the biochemical principles, instrumentation, and specific procedures.
Page 7, lines 124-135: “The serum concentrations of adiponectin, leptin, TC, TG, LDL-C and HDL-C were measured using corresponding commercial kits according to the manufacturer’s protocols. In addition, the liver concentrations of AST and ALT were also measured by the corresponding commercial kits.” was revised as “Key serum parameters were assessed as follows: TC, TG, LDL-C, and HDL-C levels were measured using corresponding enzymatic colorimetric kits (employing CHOD-PAP, GPO-PAP, and homogeneous methods, respectively) according to the manufacturer's protocols. Meanwhile, adiponectin and leptin concentrations were quantified using specific ELISA kits per the provided instructions. The assays are based on the sandwich ELISA principle, where captured antibodies specific to each analyte are coated onto the microplate. The intensity of the colorimetric signal, measured at 450 nm, is proportional to the concentration of the target hormone in the sample. A standard curve was generated for each assay to calculate the precise concentrations. The activities of AST and ALT in liver homogenates were determined using commercial assay kits following the manufacturer's protocols. These assays are based on the principle that AST and ALT catalyze specific reactions involving α-ketoglutarate and L-aspartate or L-alanine, respectively, leading to the formation of NADH, which is monitored by the increase in absorbance at 340 nm.”
- You should include a flowchart with all the steps you took during the experiments.
Response: We appreciate the reviewer's comment regarding experimental clarity. In our revised manuscript, we have significantly enhanced the structure of the methods section to ensure all steps are presented in a clear, sequential manner. Given that the experimental procedures employed (e.g., HFD feeding, oral gavage, biochemical assays, 16S rRNA sequencing, Western blot) are well-established and comprehensively described in the text, a flowchart is not conventionally included in papers of this type in the literature. We are confident that the revised, detailed methodological account fully allows for the reproduction of the study. We are, of course, open to further discussion on this point with the editor.
- The Results are well described, but the Discussion has to be improved and expanded. Its division into subsections, according to the Results section, is highly recommended. The study’s strengths and limitations need to be analyzed. A Conclusions section is missing, as well as the future perspectives and practical implications.
Response: We sincerely thank the reviewer for these excellent suggestions to enhance the depth, clarity, and critical tone of our Discussion. We have undertaken a major revision of this section.
We sincerely thank the reviewer for this constructive feedback. We have now added a dedicated "5. Conclusions" section following the Discussion. This new section succinctly summarizes the main findings of our study, explicitly outlines future research perspectives, and discusses the practical implications of our work, thereby providing a clear and impactful conclusion to the manuscript.
The newly added "Conclusions" section is presented below and has been incorporated into the main manuscript file.
Page 19, lines 377-386: “Our study demonstrates that supplementation with live EC significantly alleviates high-fat diet-induced obesity and glucolipid metabolic disorders in mice by remodeling the gut microbiota, enhancing the production of short-chain fatty acids and succinate, and subsequently activating the intestinal gluconeogenesis pathway. These findings not only elucidate a novel mechanism behind the anti-obesity effects of EC but also position it as a promising probiotic candidate for functional food applications. Future research should focus on validating these results in human clinical trials, establishing causal relationships through fecal microbiota transplantation, and identifying the key bioactive metabolites responsible for the observed benefits. This work provides a scientific foundation for developing EC-based dietary strategies to combat metabolic diseases.”
Round 2
Reviewer 1 Report
Comments and Suggestions for Authors
This paper can be accepted for publication.
Reviewer 2 Report
Comments and Suggestions for Authors
I have no further comments on this manuscript.
Reviewer 3 Report
Comments and Suggestions for Authors
Since the authors solved the high similarities with other publications and all my previous comments were properly addressed, I have no objection to suggesting this manuscript for publication.